# Hive Products: Composition, Pharmacological Properties, and Therapeutic Applications

**DOI:** 10.3390/ph17050646

**Published:** 2024-05-16

**Authors:** Roberto Bava, Fabio Castagna, Carmine Lupia, Giusi Poerio, Giovanna Liguori, Renato Lombardi, Maria Diana Naturale, Rosa Maria Bulotta, Vito Biondi, Annamaria Passantino, Domenico Britti, Giancarlo Statti, Ernesto Palma

**Affiliations:** 1Department of Health Sciences, University of Catanzaro Magna Græcia, 88100 Catanzaro, Italy; roberto.bava@unicz.it (R.B.); studiolupiacarmine@libero.it (C.L.); rosamaria.bulotta@gmail.com (R.M.B.); britti@unicz.it (D.B.); palma@unicz.it (E.P.); 2Mediterranean Ethnobotanical Conservatory, Sersale (CZ), 88054 Catanzaro, Italy; 3ATS Val Padana, Via dei Toscani, 46100 Mantova, Italy; giusi.poerio@gmail.com; 4Local Health Authority, ASL, 71121 Foggia, Italy; giovanna.liguori@aslfg.it; 5IRCCS Casa Sollievo della Sofferenza, San Giovanni Rotondo (FG), 71013 Foggia, Italy; renato.lombardi@aslfg.it; 6Ministry of Health, Directorate General for Health Programming, 00144 Rome, Italy; mariadiananaturale@gmail.com; 7Department of Veterinary Sciences, University of Messina, 98168 Messina, Italy; vbiondi@unime.it (V.B.); passanna@unime.it (A.P.); 8Department of Pharmacy, Health and Nutritional Sciences, University of Calabria, Rende, 87036 Cosenza, Italy; giancarlo.statti@unical.it; 9Center for Pharmacological Research, Food Safety, High Tech and Health (IRC-FSH), University of Catanzaro Magna Græcia, 88100 Catanzaro, Italy

**Keywords:** honey, pollen, propolis, royal jelly, bee venom, therapeutic properties, apitherapy, human and animal health

## Abstract

Beekeeping provides products with nutraceutical and pharmaceutical characteristics. These products are characterized by abundance of bioactive compounds. For different reasons, honey, royal jelly, propolis, venom, and pollen are beneficial to humans and animals and could be used as therapeutics. The pharmacological action of these products is related to many of their constituents. The main bioactive components of honey include oligosaccharides, methylglyoxal, royal jelly proteins (MRJPs), and phenolics compounds. Royal jelly contains jelleins, royalisin peptides, MRJPs, and derivatives of hydroxy-decenoic acid, particularly 10-hydroxy-2-decenoic acid (10-HDA), which possess antibacterial, anti-inflammatory, immunomodulatory, neuromodulatory, metabolic syndrome-preventing, and anti-aging properties. Propolis has a plethora of activities that are referable to compounds such as caffeic acid phenethyl ester. Peptides found in bee venom include phospholipase A2, apamin, and melittin. In addition to being vitamin-rich, bee pollen also includes unsaturated fatty acids, sterols, and phenolics compounds that express antiatherosclerotic, antidiabetic, and anti-inflammatory properties. Therefore, the constituents of hive products are particular and different. All of these constituents have been investigated for their properties in numerous research studies. This review aims to provide a thorough screening of the bioactive chemicals found in honeybee products and their beneficial biological effects. The manuscript may provide impetus to the branch of unconventional medicine that goes by the name of apitherapy.

## 1. Introduction

Hive products have important economic value. They have importance as foods and as food ingredients. In addition, they find application in numerous fields of cosmetics. These are now the main commercial directions in which bee products are conveyed. Over time, therefore, bee products have lost their importance in modern Westernized medicine [1].

However, people in developing countries, where the expense associated with the cost of drugs is often unsustainable, and followers of holistic approaches still use bee products for medical purposes. The use of bee products in medical practice is referred to as apitherapy. Apitherapy is a branch of natural, or alternative, medicine that uses beehive products to treat both serious and non-serious pathologies. It is, in fact, defined as a set of targeted treatments for the well-being and psycho-physical recovery of various organisms.

Natural products are increasingly used in the field of medicine, especially when it comes to everyday non-serious illnesses. Thanks to their beneficial and nutritional properties, beehive products could bring numerous benefits, from both a preventive and therapeutic point of view. In addition to human medicine, these natural products have found wide applications in veterinary medicine [2]. In fact, every product derived from honeybee work is a concentrate of beneficial principles that are useful for the health and well-being of the body; properties that are well known to those who study apitherapy, and which are, therefore, exploited in various applications. 

Therefore, there are several facets of apitherapy that can be principally traced back to the following macro areas: apinutrition, apicosmetics, apiaromatherapy, apitoxitherapy.

Today’s globe is seeing a rise in the popularity of biologically active bee products. Over the years, a number of honeybee products—including honey, pollen, royal jelly, propolis, beeswax, bread, and venom—have been found to be potential sources of compounds with medicinal potential [3]. Chemical heterogeneity based on the diversity of plant sources of origin prevents the clinical standardization of these products; despite this, several compounds have been described for their pharmacological properties.

Honey has been shown to be an oxidizing agent with anti-inflammatory, pro-apoptotic, anti-proliferative, anti-metastatic, and immune-modulatory qualities [4]. According to a number of studies, honey holds promise as a cancer therapy [4,5,6]. A gland in the bee’s abdomen cavity produces and secretes bee venom. The induction of cytotoxicity, necrosis, apoptosis, and proliferation inhibition in a variety of cancer cells, including those from the liver, breast, lung, prostate, and bladder, suggests that it is useful in the treatment of cancer [5]. These two examples can be complemented by several others. Honey has a vast array of uses and characteristics. It exhibits a variety of anti-inflammatory, antioxidant, and antibacterial properties. It also helps skin grafts adhere better and speeds up the healing of wounds [7]. Other important hive products are propolis, pollen, and royal jelly (RJ). Propolis is composed of several chemical substances, including amino acids, polyphenols, minerals, flavonoids, ethanol, vitamin E, vitamin B complex, and vitamin A, as well as essential oils, resins, pollen, and waxes [8,9]. It is effective against viruses and bacteria. Clinical evidence revealed that, for the treatment of genital herpes simplex, an ointment containing Canadian propolis worked better than acyclovir or placebo [3]. Additionally, propolis has the ability to stop the hepatitis C virus from replicating in vitro, limit HIV-1 activity by targeting viral integrase, and effectively combat herpes simplex 1 and 2 when combined with derivatives of caffeic acid [10,11]. Furthermore, various human pathogenic viruses, including the respiratory syncytial virus (RSV) [12], herpesviruses [13], influenza virus [14,15,16], human immunodeficiency virus (HIV) [17], human T-cell leukemia/lymphoma virus type 1 (HLTV-1) [18], Newcastle disease virus (NDV) [19], RSV [20], poliovirus (PV)–type 1 [21], and dengue virus (DENV) [22], have also been reported to be susceptible to the antiviral activities of propolis. Bee products, such propolis and honey, have recently been explored in therapeutic studies against severe-acute-respiratory-syndrome-related coronavirus 2 (SARS-CoV-2) [23]. In addition to bacteria and viruses, research has brought to light the anti-parasitic properties of hive products. Bee products have been shown to have antiprotozoal efficacy against the intestinal parasite *Giardia lamblia* [3,24,25]. Research revealed that propolis has anti-malarial properties against *Plasmodium falciparum*, *Plasmodium ovale*, *Plasmodium malariae*, and *Plasmodium vivax* [24]. As is clear from this brief overview, there are many uses and possible applications of hive products. The numerous studies on the properties of hive products are summarized in this review; in particular, we will try to highlight the significance of bee products in the medical and pharmaceutical industries. From this general overview, we can already understand that hive products have a multiorgan action that can be exploited for the general well-being of organisms (Figure 1). 

## 2. Methodology

The databases Scopus, SciELO, Web of Science, PubMed, Google Scholar, and Espacenet were used to gather literature data. We included a number of terms related to the theme of the article in the search box, such as “propolis anticancer properties”, “honey antibacterial activity”, and “hive products anti-inflammatory action”, among many others. Both the operators “AND” and “OR” were used, depending on how the words were combined. To be included in the review procedure, the study had to be published in English. Much of the important information in the text has been summarized and included in nine tables. Of these, the last one (List of Abbreviations), found at the end of the text, collects a list of abbreviations used in the manuscript.

## 3. Bee Product: Overview and Constituents

### 3.1. Honey

Honey is a food product produced by honeybees from deposits of plant and flower nectar. Foraging bees gather flower nectar and repeatedly digest and regurgitate it to generate honey. An 80% sugar solution, primarily fructose and glucose, with smaller amounts of sucrose, maltose, and other complex sugars is produced by the stomach’s acidic pH in conjunction with the enzymatic activity of invertase, diastase, and amylase [26]. The kind of honey-collecting bee, the plants from which it is harvested, the climate, the location, and the storage conditions are some of the variables that affect the chemical composition of honey [24]. Honey is a sweet substance that is primarily composed of monosaccharides, such as glucose (31%) and fructose (38%), with smaller amounts of disaccharides, such as sucrose, gentiobiose, isomaltose, kojibiose, laminaribiose, maltose, maltulose, nigerose, and trehalose, and trisaccharides, such as centose, erlose, isomaltosylglucose, isopanose, 1-ketose, maltotriose, panose, and theanderose [27,28]. Other honey components include amino acid monomers (alanine, asparagine, glutamine, glycine, and proline) and enzymes (acid phosphatase, amylase, catalase, diastase, glucose oxidase, invertase, and sucrose diastase) [27].

The most common amino acid is proline, which is followed in abundance by glutamic acid, alanine, phenylalanine, tyrosine, leucine, isoleucine, and a few other smaller ones. Honey from *Apis mellifera*, the western honeybee, typically has 0.1–1.5% of protein, while the honey from *A. cerana*, the Asiatic honeybee, typically has 0.1–3.0%. Defensin-1 and major royal jelly protein (MRJP) isoforms are the most prevalent peptides, and the main enzymes include glucose oxidase, diastase (amylase), α-glucosidase, catalase, and acid phosphatase [29]. Honey contains a variety of phenolic acids, including caffeic, cinnamic, ferulic, and others, as well as a number of organic acids, including citric and gluconic, followed by lesser quantities of acetic and formic acid [30]. The acidic pH of honey, which ranges from pH 3.4 to pH 6.1, is determined by these acids [31]. Honey also includes flavonoids such as myricetin, naringenin, apigenin, hesperetin, galangin, luteolin, quercetin, and kaempferol [30]. The honey samples’ total flavonoid content (TFC) and total phenolic content (TPC) range from 31.5 ± 2.1 to 126.6 ± 2.7 mg of gallic acid equivalents (GAE)/100 g and from 1.9 ± 0.1 to 4.2 ± 0.6 mg of quercetin equivalents (QE)/100 g, respectively [32]. Trace levels of vitamins are also present, particularly the vitamin B complex from pollen grains. All of the water-soluble vitamins and minerals such as P, Na, Ca, K, S, Mg, Cl, Si, Rb, V, Zr, Li, and Sr are recorded in very low quantities [28,33]. Furthermore, volatile substances such alcohols, aldehydes, benzene and its derivatives, terpene and its derivatives, ketones, pyran, furan, and acid esters have been found in honey [7,34]. The main bioactive components of honey include oligosaccharides, methylglyoxal, royal jelly proteins (MRJPs) and phenols [30,35]. A better understanding of the health benefits of honey, including its anticancer, antiallergic, antibacterial, antioxidant, antidiabetic, antiparasitic, antiulcer, anti-inflammatory, wound-healing, and cardioprotective qualities, has been made possible by the existence of this varied chemical composition. The main components of honey are listed in Table 1.

### 3.2. Propolis

Honeybees generate propolis by combining the exudate from their salivary glands with plant exudate that has been collected from various plant components, including branches, bark, flower buds, leaves, and stems. Propolis is a hard, brittle substance that is lipophilic in nature. It comes in a variety of colors, from green to brown and reddish. 

When propolis is heated, it becomes soft and sticky and has a pleasant or sweet smell [48]. Propolis’s chemical makeup is significantly influenced by its floral and geographic origin. More than 300 distinct chemicals are typically present in raw propolis, with the majority being triterpenes (50 percent *w*/*w*), waxes (25–30%), volatile mono- and sesquiterpenes (5–8%), and phenolics (5–10%) [8]. Triterpenes are responsible for propolis’s characteristic resinous odor. Raw propolis typically contains 50–60% resins and balms, 30–40% fatty acids and waxes, 5–10% essential oils, and the remaining 5–10% are made up other components, which include vitamins B1, B2, B6, C, and E, minerals (Mg, Cu, F, Ca, K, Na, Mn, and Zn), and enzymes (acid phosphatase, adenosine triphosphatase, glucose-6-phosphatase, and succinic dehydrogenase) [8,49]. In order to eliminate inert components and maintain the phenolic fractions for commercialization, propolis has to be purified and dewaxed via solvent extraction [50]. Propolis contains a number of monosaccharides, including glucose and fructose, as well as a disaccharide, sucrose. Additionally, it contains phenolic acids (caffeic acid, chlorogenic acid, cinnamic acid, gallic acid, 4-hydroxybenzoic acid, 4-hydroxyhydrocinnamic acid, and 4-hydroxybenzoic acid-methyl ester), terpenoids, and flavonoids (apigenin, chrysin, acacetin, catechin, daidzein, formononetin, naringenin, galangin, kaempferol, luteolin, liquiritigenin, myricetin, pinocembrin, rutin, and quercetin) [8]. Brazilian, Chinese, and Australian propolis are among the samples in which the TPC and TFC vary from 127–142 mg GAE/g and 33–53 mg QE/g, respectively [51]. Propolis also contains fatty acids [52], including arachidonic, cis-13, 16 docosadienoic, cis-11,14,17-eicosatrienoic, cis-5,8,11,14,17-eicosapentaenoic, eicosadienoic, elaidic, heneicosylic, linoleic, oleic, palmitic, and palmitoleic acid. Table 2 shows important constituents of propolis.

### 3.3. Pollen

Worker honeybees generate bee pollen, which serves as the larvae’s main source of nutrition [64,65]. Bee pollen is the result of combining components of honeybee salivary enzymes with floral nectar and flower pollen. The types of plants, bee activity, and meteorological factors all affect the chemical composition of bee pollen [66]. Bee pollen comes in a variety of forms and colors. The single constituent grains can be bell-shaped, cylindrical, thorny, and triangular. The grains are combined with two or more additional grains to make up bee pollen [67]. 

Honey, proteins, amino acids, fats and oils, phenolics, enzymes and coenzymes, vitamins, and minerals are the principal chemical components of bee pollen [67]. However, the biological qualities and therapeutic benefits of bee pollen are greatly influenced by its extremely changeable chemical makeup, which varies greatly depending on the plant source, geographic location, and climatic circumstances [68].

Bee pollen contains a high percentage of carbohydrates (35–61%), with monosaccharides (fructose—24% and glucose—11%), disaccharides (4–9% sucrose), and other sugars (1–2%) including raffinose, erlose, isomaltose, maltose, melibiose, melezitose, rhamnose, ribose, trehalose, and turanose [66,69,70]. Furthermore, 14–30% of bee pollen is protein, and 10.4% of that protein is made up of important amino acids, which include valine, phenylalanine, tryptophan, histidine, isoleucine, leucine, lysine, and methionine [67,70,71]. Similarly, bee pollen has lipids in larger proportions (between 1 and 13%) compared to proteins and carbs. Myristic, palmitic, and stearic acids make up the majority of saturated fatty acids (4.3–71.5%), while oleic, linolenic, and linoleic acids make up the majority of unsaturated fatty acids (1.3–53.2%). Bee pollen also contains arachidonic, behenic, capric, caproic, caprylic, 11-eicosenoic, eicosatrienoic, elaidic, lauric, and lignoceric acids [65,72]. Moreover, it has been shown that bee pollen contains tannins, phenolic acids, and flavonoids [65]. The total flavonoid content (TFC) and total phenolic content (TPC) of bee pollen from various countries are, respectively, 1.00–5.50 mg QE/g and 0.50–213 mg GAE/g [73]. About 1.4% of the pollen from bees contains the major flavonoids, which are isorhamnetin, kaempferol, quercetin, and its 3-O-glucosides; these are followed by luteolin, naringenin, apigenin, epicatechin, hesperetin, and catechin [67,74]. The following phenolic acids are found in bee pollen: rosmarinic, syringic, caffeic, chlorogenic, ferulic, gallic, p-coumaric, p-hydroxybenzoic, protocatechuic, and vanillic acid [65]. Vitamins can be found in bee pollen: 0.6% of them are water soluble, including vitamin B1, B2, B3, B5, B7, B6, B8, B9, C, and vitamin P; 0.1% of them are fat soluble, like provitamin A (beta-carotene), vitamin E, and vitamin D [75]. Minerals are among the beneficial compounds found in bee pollen, together with micronutrients (Fe, Cu, Cr, Mn, Se, Si, Zn) and macronutrients (Ca, K, Mg, Na, P) [76]. Table 3 below lists the main components of pollen.

### 3.4. Royal Jelly

The mandibular and hypopharyngeal glands of worker honeybees produce a thick, white-to-yellow jelly-like fluid known as royal jelly, or bee’s milk [78,79]. It has a pH of 3.1–3.9, a pungent smell, and a sour or sweet taste. It is also mildly soluble in water [80]. A significant part of the nutrition of honeybee larvae is royal jelly. It is only supplied to worker and young drone larvae throughout their maturation phase, and it is fed to queen honeybees for the duration of their life cycle [81]. Water makes up 50–70% of royal jelly, followed by carbohydrates (30%), proteins (27–41%), and lipids (3–19%). Royal jelly primarily contains two types of sugars: fructose and glucose. Furthermore, extremely minute amounts of sucrose and other oligosaccharides such as erlose, gentobiose, isomaltose, maltose, melezitose, raffinose, and trehalose are found [82,83]. Nine distinct soluble main royal jelly proteins (MRJPs 1–9) serve as the particular elements that are in charge of the development of queen honeybees. It has been shown that the peptides found in royal jelly, such as apisimin, jelleines, and royalisin, have antibacterial properties [3]. Fatty acids (80–85%), waxes (5–6%), steroids (3–4%), and phospholipids (0.4–0.8%) are listed as the constituents of the lipid composition. The fatty acids found in royal jelly are typically either rare short-chain hydroxy or dicarboxylic acids (8–12 carbon atoms) such as trans-2-decenoic acid, 10-hydroxy-trans-2-decenoic acid (10-HDA), and 10-hydroxydecanoic acid (HDAA), and sebacic acid, 3-hydroxydecanoic, 9-hydroxy-2-decenoic, 8-hydroxyoctanoic, and 9-hydroxydecanoic acid. Among them, unique royal jelly components include 10-HDA and 10-HDAA [84,85,86]. Flavanones (pinobaskin, pinocembrin, hesperidin, naringin, and naringenin), flavones (acacetin, apigenin, chrysin, and luteolin), flavonols (fisetin, galangin, isorhamnetin, kaempferol, quercetin, and rutin), and phenolic acids (caffeic acid, gallic acid, 4-hydroxy-3-methoxyphenylethanol, 4-hydroxybenzoic acid-methyl ester, 4-hydroxybenzoic acid, 4-hydroxyhydrocinnamic acid) as well as other phenolic compounds, are represented [48,76]. The ranges of the TPC and TFC in royal jelly are 3 to 9 mg GAE/g and 0.1 to 0.5 mg QE/g, respectively [80]. Royal jelly also includes minerals (Cu, Fe, K, Mg, and Zn), vitamins (B1, B2, B3, B5, B6, B9, and provitamin A), and hormones (prolactin, testosterone, estrogen, and progesterone) [85]. Table 4 shows the important components of royal jelly.

### 3.5. Bee Venom

A gland in the bees’ abdominal cavity secretes apitoxin, or bee venom. Bees often use this clear, acidic liquid with no smell as a weapon in their defense against potential predators. The venom of honeybees is a mixture of several substances. Numerous publications have reported that bee venom contains a variety of active molecules, including peptides and enzymes, such as phospholipase A2 and hyaluronidase, as well as non-peptide components like histamine, dopamine, and norepinephrine [2]. Melittin, a major component of bee venom, is also present, as are apamin, adolapin, and mast cell degranulating peptide. Phospholipase A2 (PLA2), which makes up around 12 percent of the venom, and melittin, which makes up about 50 percent, are the principal ingredients [90].

Bee venom and bee-derived toxins have been utilized in traditional medicine to treat chronic inflammatory illnesses because of their many effects, including their anti-arthritic, anti-cancer, and analgesic properties [90,91,92]. In bee venom treatment, lyophilized venom (which is extracted from bees and subsequently lyophilized) is injected directly by various dosages in situ, but in bee sting therapy, the honeybees go directly to the target location via their stinger [93]. A variety of ailments, including autoimmune disorders (rheumatoid arthritis, psoriasis, and so forth), neurological disorders, chronic inflammations, pain, skin conditions, and microbiological infections may be treated by injecting bee venom [94]. Table 5 lists the important components of bee venom.

## 4. General Pharmaceutical Properties of Hive Products

### 4.1. Anti-Inflammatory Properties

The production of pro-inflammatory TNF-α and IL-1β from LPS-stimulated N13 microglia cells has been reduced by an extract from Italian multifloral honey that contains the flavonoids daidzein, apigenin, genistin, luteolin, kaempferol, quercetin, and chrysin as main components [96]. These results support the potential use of the honey flavonoid fraction in several illnesses such as Parkinson’s disease and Alzheimer’s disease, given the significance of neuroinflammation in neurodegenerative diseases. 

Additionally, honey proteins have been shown to have immunomodulatory properties. Antigen-stimulated T cells have been shown to produce less IFN-γ, IL-2, and IL-4 when exposed to MRJP-3 [97]. Ziziphus honey contains glycopeptides and glycoproteins ranging in size from 2 to 450 kDa. These compounds have been shown to inhibit the release of ROS by human neutrophils and murine macrophages activated by zymosan, the production of NO and phagocytosis by LPS-activated murine macrophages, and the production of TNF-α by human monocytic cells [98].

It has been shown that the honey protein apalbumin-1, also known as MRJP-1, blocks the mannose receptors of human phagocytic cells, preventing phagocytic activity. Because of apalbumin glycation, this inhibitory action seems to be amplified in honey containing MGO [99].

It was found that 10-HDA can shield rats from artificially created stomach ulcers in a study on the potential of royal jelly for digestive trait illnesses [100]. The suppression of LPS-induced NF-κB activation shown in the murine macrophage cell line RAW264 is one mechanism suggested to be connected to the anti-inflammatory activity of 10-HDA [101]. It has been shown that 10-HDA and 4-hydroperoxy-2-decenoic acid ethyl ester suppress histone deacetylase activity, which increases leukemia THP-1 cells’ expression of extracellular SOD release and suggests that these compounds may have therapeutic potential against atherosclerosis [102]. It is believed that 10-HDA inhibits histone deacetylase, which in turn causes epigenetically suppressed genes to reactivate in mammalian cells. This has led to the theory that caste flipping in bees may be caused by a similar mechanism [103]. A study demonstrating 10-’ad’s suppression of fibroblast-like synoviocytes from rheumatoid arthritis patients also revealed modifications in histone acetylation, pointing to its possible therapeutic benefits against chronic inflammatory degenerative illnesses [104].

Numerous studies have shown propolis’s anti-inflammatory qualities, which may be related to the compound’s phenolic acid content. According to Armutcu et al. (2015) [105], CAPE is thought to be a very potent anti-inflammatory ingredient because it may target NF-κB signaling directly. Additionally, it has been shown that this substance regulates the PI3K/Akt pathway in many human cell lines [106] and modulates ERK MAPK signaling in T cells and mastocytes [107]. The downregulation of important inflammatory enzymes such as xanthine oxidase, cyclooxygenase, matrix metalloproteinases, and inducible nitric oxide synthase might be a potential consequence of these anti-inflammatory processes [105,106]. 

Propolis’s anti-inflammatory properties are often included in mouthwash formulations. Phenolics, particularly CAPE, have been linked to antigingivitis action [106]. Furthermore, rinse solutions that are high in artepillin C from Brazilian green propolis have been shown to reduce gingivitis to an equivalent degree as either a chlorhexidine solution or a NaF/cetylpyridinium chloride rinse in randomized, double-blind, placebo-controlled studies [108]. Applying propolis topically also makes the skin more lenitive. Propolis from Romania and Australia has shown photo-protective effects in animal models, maybe because polyphenols contain anti-UV qualities [109,110].

According to Middleton (1998) [111] and Choi (2007) [112], bee pollen has been found to have anti-inflammatory properties comparable to those of common non-steroidal anti-inflammatory drugs. This effect may be attributed to the activity of flavonoids, phenolic acids, phytosterols, and flavoring substances such as anethole, which is an inhibitor of the NF-KB pathway. According to Yakusheva (2010) [113], some specific effects include the capacity to reduce swellings brought on by renal and cardiovascular diseases, shield the liver from damage caused by carbon tetrachloride [114], and lessen inflammation and hyperplasia in the prostate. Antiandrogen activities have also been linked to positive effects on prostatic diseases [115].

There are at least four anti-inflammatory compounds found in bee venom. Phospholipase A2, adolapin, melittin, and apamin are a few of the most significant. Mellitin’s anti-inflammatory effects against neuroinflammation, amyotrophic lateral sclerosis, atherosclerosis, and liver inflammation have been the subject of much research [116,117,118,119]. MAPKs (mitogen-activated protein kinases), such as p38 MAPK and ERK, are markedly inhibited by melittin [120]. The body has a large number of enzymes called MAPKs (mitogen-activated protein kinases), which are involved in many physiological and pathological processes. p38, in particular, is a protein that is implicated in a variety of cellular responses, making it an extremely intriguing pharmaceutical target. Its enzymatic activity takes place in the nucleus as well as the cytoplasm. Numerous cellular functions, including mRNA regulation, apoptosis, protein degradation, and chromatin and cytoskeleton organization, are impacted by it. Studies conducted both in vivo and in vitro have linked inflammation to the activation of p38, namely p38α, which has been shown to interfere at both transcriptional and post-transcriptional levels. It activates COX-2 and controls the synthesis of many proinflammatory cytokines (IL-1β, IL-6, TNF-α, and INF-gamma), which are essential in the development of asthma, COPD (chronic obstructive pulmonary disease), and autoimmune diseases (like multiple sclerosis, rheumatoid arthritis, and Crohn’s disease) as well as cardiovascular conditions like atherosclerosis [121,122,123].

Consequently, melittin therapies inhibit the activation of the TLR system and stop the production of inflammatory cytokines [124]. Melittin can inhibit MAPK serine p38 inhibition and nuclear NF-kB p65 activation in vitro [125]. Thus, this function has anti-inflammatory properties. Melittin has been shown to have anti-inflammatory properties by altering the transcription factors NF-kB and AP-1 in vivo [126].

### 4.2. Bee Products’ Anticancer Properties

Honey has anticancer properties since it exhibits cytotoxicity against several cancer cell types. For example, honey samples from Morocco reduced the cell viability of human colorectal cancer (HCT-1) cell cultures in an MTT experiment [127]. Phenolic substances including rosmarinic acid, tannic acid, caffeic acid, coumaric acid, gallic acid, ferulic acid, syringic acid, catechin, and pyrogallol were found in the active honey samples [127]. At different doses, Manuka honey has been shown to effectively suppress the proliferation of MCF-7 (a breast cancer cell line) [128,129,130,131]. At a concentration of 5.5% *v*/*v*, acacia honey also demonstrated anticancer action against MCF-7 [132]. In addition to HCT-1 and MCF-7, the development of PC-3, a prostate cancer cell type, was also shown to be inhibited by honey (0.5 to 1 mg/mL) [133]. Human breast cancer cell lines (MCF-7 and MDA-MB-231) and cervical cancer cell lines (HeLa) were also susceptible to tualang honey (TH) action. TH is cytotoxic to the two types of cancer cells at effective concentrations (EC50) of 2.4–2.8%, as evidenced by the enhanced lactate dehydrogenase (LDH) leakage from cell membranes [134]. 

Numerous components of honey have undergone separate testing to determine their antiproliferative properties. One of the most well researched phenolics in honey is chrysin, which has been shown to have harmful effects on a number of cancer cell lines. Chrysin, for instance, was shown to inhibit human melanoma (A375) cell lines by 15% and 25% at doses of 25 µM and 50 µM, and to inhibit murine melanoma (B16-F1) cell lines by 10% and 20% after 24 h of therapy. The MTT test was used to obtain these findings [48]. After 72 h of incubation, a half-maximal inhibitory concentration (IC50) of 50 µM was discovered for both human and murine melanoma cell lines. In the same investigation, acacia honey inhibited both human and murine melanoma cell lines in a time- and dose-dependent manner, with an estimated IC50 value of around 0.02 g/mL [135]. Chrysin, however, was shown to maximally reduce HCT16 cell viability in human colon cancer cell lines at a final concentration of 100 µM, with about 13% inhibition seen after 6 h and approximately 78% inhibition after 48 h of incubation [135]. It is interesting to note that the control cell lines were unaffected by the toxic effects, which were limited to the cancer cells. Furthermore, chrysin’s cytotoxicity has been documented against a number of different cancer cell lines, such as those from the breast, prostate, cervix, liver, glioblastoma, lung, and pancreas [135,136,137]. 

Quercetin has been shown to have antiproliferative effects against HL-60 leukemia [138], MCF-7 human breast cancer [139], Caco-2 human colon adenocarcinoma [140], PC3 and DU145 prostate cancers [141], SCC25 oral cancer [142], Ishiwa endometrial cancer [143], and SPC212 and SPC111 malignant mesothelioma cell lines [144]. Concentrations as low as 10 µM were reported to decrease cancer cell development in the HL-60 cell line by 17% after 24 h and by about 53% after 96 h of incubation [138].

A quinoline alkaloid was shown to be in charge of the apoptotic mechanism during the assessment of chestnut honey’s anticancer efficacy [145]. Additional honey constituents that prevent the growth of cancer cells include antioxidant components, which protects cells from free radical damage. Furthermore, honey has the ability to induce apoptosis through cellular signaling pathways that modulate immune activity [129,145,146,147]. 

Propolis’s antiproliferative properties have also been the subject of much research in recent years. Because of its resinous nature, propolis is often extracted using methanol, ethanol, or other organic solvents before its pharmacological effects are assessed; unlike honey, which is studied in its unprocessed form. Propolis extract from Turkey showed the ability to inhibit cell growth in cytotoxic tests conducted on A549 cells, a model of human lung cancer cells [148]. It has been shown that the ethyl acetate fraction of Saudi Arabian propolis inhibits human liver cancer cells (HEP-62) and squamous carcinoma cell lines (SW-756) as well as Jurkat cells, a T-lymphocyte leukemia model [149]. Likewise, it has been observed that Lebanese propolis inhibits the proliferation of Jurkat cells [150]. It is interesting to note that, in contrast to the aqueous and dichloromethane fractions, only the hexane fraction exhibits inhibition against additional carcinoma cell types, including U251 (glioblastoma) and MDA-MB-231 (breast adenocarcinoma) [150]. These findings imply that the anticancer properties of propolis may be attributed to less polar components. Among the most often reported phytochemicals in propolis are simple polyphenol compounds such caffeic acid, chrysin, p-coumaric acid, galangin, ferulic acid, and pinocembrin. It has also been proposed that these substances are important in inhibiting the proliferation of cancer cells. Czyzewska used CAL-27 cells, a human tongue squamous carcinoma model, to evaluate the anticancer efficacy of individual components (chrysin, galangin, caffeic acid, and p-coumaric acid) of ethanolic extract of propolis compared to a combination of these polyphenols. Flow cytometry reveals that the mechanism of cytotoxicity mediated by these components is apoptosis. The propolis ethanolic extracts triggered caspases-3, -8, and -9. The most effective approach to trigger apoptosis through both intrinsic and extrinsic pathways was discovered to be a mixture of polyphenols. It is crucial to remember that the mixture of polyphenolic compounds was tested at a higher concentration of polyphenols rather than simulating the relative concentration of each substance found in the propolis that was tested [151]. Indirect mechanisms of action are also implicated in the antitumor activity of propolis. Propolis may prevent the growth of cancer by strengthening the immune system, along with other potential ways. The impact of propolis’s polyphenolic chemicals and water-soluble derivatives (WSDP) on the formation of Ehrlich ascites tumors (EAT) in mice was studied by Oršolić et al. (2005) [152]. A tumor was created by injecting 2 × 10^6^ EAT cells into the peritoneal cavity. The mice were administered WSDP plus three polyphenolic compounds by oral (po) administration: quercetin (QU), caffeic acid (CA), and caffeic acid phenethyl ester (CAPE). Treating EAT-bearing mice with test components resulted in a significant reduction in the volume of ascitic fluid produced by EAT cells and the overall number of cells in the peritoneal cavity, and led to an extended life period for the animals. It was their impact on the mice’s immune systems that inhibited the growth of EAT. A dose-related increase in the activity of cytotoxic T-cells, natural killer cells, and B cells was seen in mice treated with test components when innate and acquired immune responses were assessed [152]. Propolis samples from northern Morocco have been shown to be cytotoxic against MCF-7, HCT, and THP-1. They have also been shown to increase the production of interleukin-10 (IL-10) and decrease the production of TNF- and IL-6 [153], indicating that the propolis may have immunomodulatory properties that could aid in the fight against the tested cancer cells. According to predictions, propolis’s other anticancer activities involve interactions with microtubules and the induction of tubulin depolymerization [149], caspase-3, -8, and -9 activation of apoptosis [151], and proline dehydrogenase/proline oxidase activity-induced reductions in proline in cancer cells [154]. 

Additionally, propolis may impair the efficiency of cytotoxic drug-based chemotherapy. In a mouse model of colorectal cancer, Sameni et al. (2021) demonstrated that administering Iranian propolis extract in conjunction with 5-fluorouracil (5-FU) significantly decreased the frequency of azaxymethane-induced aberrant crypt foci in comparison to propolis or 5-FU alone. Additionally, the propolis and 5-FU combination reduced the expression of the proteins Cox-2, iNOS, and β-catenin, which are crucial for the development and occurrence of colorectal cancer [155]. Brazilian green propolis contains a phenolic chemical called artepillin C (Art-C), which is a prenylated derivative of p-coumaric acid [156,157]. Strong anticancer activity is exhibited by artepillin C against a variety of cancer cells. In gastrointestinal cancer cell lines, Akao and colleagues (2003) [158] demonstrated that the active components in propolis had an inhibitory effect on cell growth. In this investigation, all cell lines’ proliferation was significantly suppressed when they were exposed to 150 μM Art-C, and this impact was more pronounced than it was with other cinnamic acid derivatives found in propolis, such as baccarin and drupanin. Another study revealed that, via triggering G0/G1 arrest, Art-C extracted from Brazilian propolis suppressed cell growth in a dose-dependent manner. This could be the result of downregulating the activity of the cyclin proteins cyclin D and cyclin-dependent kinase 4 (D/CDK4) and upregulating Cip1/p21, a protein CDK inhibitor, following a 12 h therapy [159]. The proteins cyclin D and cyclin-dependent kinase 4, or Cdk4, a serine-threonine kinase, combine to form the multiprotein structure known as the Cyclin D/Cdk4 complex. The cyclin/cyclin-dependent kinase complexes, which control the cell cycle and its advancement, are the “hearts of the cell-cycle control system [160]. In castration-resistant prostate cancer (CRPC) CWR22Rv1 cells, this chemical promotes apoptosis as shown by DNA fragmentation and elevations in cleaved caspase-3 and poly ADP-ribose polymerase [161]. Szliszka et al. (2012) [162] demonstrated how artepillin C modulated the TRAIL-mediated (tumor necrosis factor-related apoptosis ligand inducer) apoptotic signaling pathways, a powerful inducer of death in cancer cells, and lowered NF-κB activity to prevent cancer in LNCaP prostate cancer cells.

One of propolis’s active ingredients is a caffeic acid phenethyl ester (CAPE). Propolis’s most researched constituent, known as CAPE, is assumed to be the source of its many biological actions in in vitro experiments. Oršolić et al.’s (2004) study [163] shed light on its function in propolis’s anticancer effects by showing that exposure to CAPE increased the rates of apoptosis in fibrosarcoma cell lines by up to 31.24%. Furthermore, Lee et al. (2000) [164] found that oral squamous cell carcinoma and submucous fibrosis-derived fibroblasts were significantly resistant to analogues of CAPE [134]. Furthermore, a number of in vitro studies have shown that CAPE has a cytotoxic effect on a variety of cancer cell lines. These studies include Chen et al. (cell death in leukemic cell lines) [165], Lee et al. (p53 and p38 MAP-kinases in cell death) [164], Hung et al. (cell death in cervical cancer cell line) [166], Jin et al. (mitochondria-mediated cell death in leukemic cell lines) [167], and Watabe et al. (NF-κB inhibition and Fas activation in breast cancer cells) [168]. It acts as an apoptotic promoter via the caspase-3/7 pathway. Through positive regulation of DR5 (death receptor 5), mediated by CHOP (C/EBP family transcription factor), CAPE dramatically boosted the amount of apoptosis mediated by TRAIL [169,170,171]. By producing more reactive oxygen species (ROS) and reducing the expression of apoptosis inhibitors like the X-linked inhibitor of apoptosis protein (XIAP), the baculoviral IAP repeat-containing protein 3 (cIAP-2), and the transporter associated with antigen processing 1 (cTAP-1) [172,173], CAPE also has an impact on the apoptotic intrinsic pathway. Apoptosis is also induced by CAPE, as evidenced by research on the reduction of carcinogenesis-associated proteins such as Akt, glycogen synthase kinase 3 beta (GSK3b), class O forkhead box transcription factor 1 (FOXO1), FOXO3a, NF-kB, S-phase kinase-associated protein 2 (Skp2), and cyclin D1 [174,175,176].

Propolis contains chrysin, a substance whose pro-apoptotic properties and molecular mechanism are well known. It has been demonstrated that this substance triggers apoptosis through the mitochondrial route. Chrysin was discovered to increase cytoplasmic Ca^2+^ levels, lipid peroxidation, and the generation of reactive oxygen species (ROS) while decreasing mitochondria membrane potential (MMP) [177].

Protein kinase B and the mammalian target of rapamycin (mTOR) were phosphorylated less when chrysin produced reactive oxygen species (ROS). Chrysin also activated unfolded protein response proteins (UPR), including the 78 kDa glucose-regulated protein (GRP78), the eukaryotic translation initiation factor 2α (eIF2α), and the PRKR-like ER kinase (PERK), to cause endoplasmic reticulum (ER) stress. Additionally, studies show that chrysin-induced apoptosis involves the PI3K/Akt and MAPK pathways [173,178,179,180].

Galangin, another propolis ingredient whose molecular mechanism of its pro-apoptotic effects has been studied, dramatically and dose-dependently decreased autophagy and apoptosis in mice harboring B16F1 melanoma tumor cells [181]. Galangin may have a suppressive effect by blocking the PI3K/AKT signaling pathway and lowering the expression of Bcl-2 protein in human laryngeal cancer cell lines [182]. Galangin also enhances TRAIL (death receptor)-mediated apoptosis and causes cell apoptosis through the activation of p38 mitogen-activated protein kinase (p38 MAPK) [173,183]. 

Another bee product whose anticancer properties have been studied is bee pollen. In contrast to other bee products, bee pollen seems to have a comparatively lower anticancer effect. Even at 100 µg/mL of bee pollen, the survival of cultured cells was not reduced in an in vitro test of anticancer activity employing mouse B16 melanoma cells [184]. 

On the other hand, it suppresses intracellular tyrosinase (TYR) and the tyrosinase receptors’ (TRP-1 and TRP-2) production of mRNA corresponding to TYR [184]. Tests on human prostate cancer (PC-3), human lung carcinoma (NCI-H727 and A549), MCF-7, and AGS yielded IC50 values ranging from 0.9 to >25 mg/mL for bee pollens taken from various locations in South Korea [185]. Studies examining how pollen extracts from *Brassica campestris* L. bees affect the viability of human prostate cancer cells have shown that the sterol fraction of a chloroform extract significantly boosts the activity of the enzyme caspase-3 and reduces the expression of the anti-apoptotic protein Bcl-2. As a consequence, human androgen-independent prostate cancer PC-3 cells, which causes programmed cell death. According to the results, there is hope for treating advanced prostate cancer using the steroid fraction of the *Brassica campestris* L. bee pollen chloroform extract [186].

The proteins from bee pollen that have been enzymatically cleaved—also referred to as hydrolysates—have shown stronger anticancer activities. The human bronchogenic carcinoma model ChaGo-K1 cells were shown to be inhibited by hydrolyzed peptides with a molecular weight of less than 65 kDa and an IC50 of 1.37 µg/mL [187]. It is clear from the aforementioned findings that, in order to inhibit certain cancer cell lines, larger amounts of bee pollen are needed. Although pollen does not have pronounced anticarcinogenic properties, it may prevent the onset of cancerous forms through the modulation of oxidative stress and inflammatory response. Exogenous dietary antioxidants have been shown to reduce oxidative stress in several trials [188,189]. 

The ability of bee pollen biocompounds to affect cellular signaling pathways, through the phosphorylation of specific proteins, may be another method of significant influence on cellular function [190,191]. 

Bee pollen flavorings, such as anethole, are known to be strong inhibitors of the activation of nuclear factor (NF)-*κ*B, which is triggered by the tumor necrosis factor (TNF). Because proinflammatory genes, such as those encoding cytokines and adhesion molecules, are expressed, the nuclear factor (NF)-ΝB pathway has been identified as a proinflammatory signaling route. Thus, the proinflammatory NF-*κ*B pathway is inhibited, and bee pollen’s anti-inflammatory properties are expressed [111]. Additionally, beebread has been evaluated against non-small cell lung cancer (NCI-H460), HeLa, MCF-7, and HepG-2, although the potency was only moderate to low (GI25 > 400 to 68 µµg/mL) [192]. Its primary constituents are polyunsaturated and monounsaturated fatty acids [170]. However, its flavonoids and polyphenolic constituents, such as isorhamnetin-O-glycoside, quercetin-O-glycoside, herbacetin glycosides, kaempferol, and myricetin [192], are believed to be the cause of its anticancer potency.

There have also been reports of the anticancer effects of bee venom (BV) [193,194]. Melittin, a significant protein present in the venom of the majority of bee species belonging to the Apis genus, is one of the most remarkable elements of bee venom. Melittin is the component of BV that has the strongest cytotoxic effect on cancer cells. An initial study demonstrating melittin’s anticancer effect found that leukemic cells inhibited calmodulin, which led to the death of cancer cells. The inhibition of the Ca^2+^ channel pump resulted in a significant rise in the Ca^2+^ concentration, ultimately leading to the death of cells [195]. Since then, several studies employing different kinds of tumor cell lines have been conducted to examine the anticancer effects of melittin and their methods of action.

During carcinogenesis, a variety of growth factor receptors, including the TNF receptor and epidermal growth factor receptor, are activated at the cell surface. The activation of these receptors sets off a number of downstream signaling cascades. The pathways that bee venom components target are the important Ras/mitogen-activated protein kinase (Ras-MAPK) pathway, which includes the Extracellular signal-regulated kinase (ERK) and Jun N-terminal kinase (JNK) routes, the phosphatidylinositol 3′ kinase (PI3K)/AKT pathways, Phospholipase C-γ (PLC-γ-CaM), and the Nuclear factor kappa-light-chain-enhancer of activated B cells (NF-kB) pathways. Phosphatidylinositol (3,4)-bisphosphate (PtdIns(3,4)P2) and bee venom-soluble phospholipase A2 (bv-sPLA2) are responsible for this cytotoxic action because they cause cell death due to membrane integrity loss, the absence of signal transmission, and the production of cytotoxic lysophosphatidylinositol 3,4-bisphosphate (lyso-PtdIns(3,4)P2). Certain components of bee venom inhibit receptors on the surface. The inhibition of their activity can be achieved either by dephosphorylation of the receptor or by its direct destruction; this, in turn, modifies the signaling pathways downstream that are crucial for proliferation, metastasis, angiogenesis, and apoptosis (for example, the synergistic effect of BV sPLA2 and PtdIns(3,4)P2). The adjustment of proliferation, differentiation, metabolism, and apoptosis is achieved through the failure to activate the Pathways Akt and Erk. The inhibition of the NF-KB reporting waterfall is another common target of bees’ poison components. In addition, the reactive species of oxygen (ROS), induced by different components of the bee poison, activate the members of the p53 family, which promotes cell cycle arrest [196]. Bee venom increases the expression and levels of different pro and apoptotic mediators, while reducing the Bcl-2 anti-apoptotic mediator. These mediators include cytochrome C (Cyt C), protein 53 (p53), Bcl-2-associated X protein (Bax), Bcl-2 homologues antagonist/killer (Bak), caspase-3, caspase-9, and various death receptors [196,197,198]. The bee venom’s anticancer action involves other pathways. Bee venom inhibits the production of metalloproteinases, which are enzymes involved in the degradation of collagen and, consequently, in pathological invasion processes, such as tumor metastasis. By inhibiting the NF-KB pathways and extracellular/mitogen-activated protein kinase p38 (ERK/p38 MAP) regulated protein kinases, matrix metalloproteinase-9’s (MMP-9) expression and activity can be reduced [178,199]. When tested against A375 (human malignant melanoma), melittin from *Apis florea* and *Apis mellifera* has been shown to display a comparatively significant anticancer activity (IC50 = 3.38 and 4.97 µg/mL, respectively), similar to that of doxorubicin [200]. Melittin’s anticancer properties were shown in a cytotoxicity test against HeLa, WiDr, and Vero cell lines, with IC50 values of 2.54, 2.68, and 3.53 µg/mL, respectively [192]. With an IC50 of 6.25 µg/mL, melittin also exhibits cytotoxic action against the human breast cancer cell line MDA-MDB-231 [195]. 

Melittin may decrease the viability of cultivated AGS cells, a model of gastric cancer, at a dosage of 0.5 µg/mL [196]. Melittin’s potential anticancer effect may be due to its ability to initiate the apoptotic pathway through the release of cytochrome-c, leading to the activation of caspase-9, which in turn triggers caspase-3 [200]. Further research on this topic revealed that the transmembrane protein EGFR’s expression is directly inhibited by melittin. It is well known that EGFR binds to the cytoskeleton of F-actin. This contact sets off downstream events that are connected to cell invasion, survival, proliferation, and metastasis through the EGFR signal transduction pathway [200]. Therefore, melittin seems to be a potential anticancer drug, However, its potential could be limited by a possible cytotoxic action against normal cells. 

Bee venom is also noted for its harmful cytolytic effects in general. Thus, efforts have been made to prevent or reduce the negative effects of administering bee venom in cancer treatment. Using specialized drug delivery vehicles, such as nanoparticles, to transport the toxin protein [197,198] and conjugating the toxin to certain cancer-targeting biomolecules [5,199,201] are a few of the answers to this issue. 

The primary compound in royal jelly that is thought to be responsible for its anticancer activity is called 10-hydroxydecenoic acid (10-HDA); it is found only in royal jelly (among the other bee products) at relatively high concentrations [202]. But according to a different study, human colorectal cancer (Caco-2) cells could not be inhibited from growing by royal jelly or 10-HDA alone; instead, a combination of royal jelly and human IFN-3N at a 2:1 ratio dramatically decreased the vitality of the cells [203]. Miyata et al. (2020) conducted an additional study to evaluate the anticancer efficacy of royal jelly using a double-blind, randomized clinical trial. While royal jelly’s anticancer efficacy was shown to be negligible, patients who received royal jelly as an adjuvant for tyrosine kinase inhibitors had a decrease in the frequency of adverse events [204,205]. However, despite royal jelly having a certain ability to preserve patients’ renal functions, Osama et al. (2017) revealed that this potency was minor when it came to cisplatin’s anticancer treatment [206]. In addition, research on the mechanisms behind royal jelly’s anticancer action has shown that it may boost mononuclear cell cytokine production in order to inhibit the development of the leukemia cell model U937 [207]. 

Other anticancer activity pathways have been proposed. For example, Bincoletto et al. (2005) [208] have shown that royal jelly has immunomodulant properties, attributable to the reduction of prostaglandin E2. Mice with Ehrlich’s ascites tumor (EAT), a tumor that impedes splenic hemopoiesis and causes immunological and hematological dysfunction as well as a decrease in the number of granulocyte-macrophage colonies, were given RJ by the study’s authors. It was found that royal jelly reversed the splenic hematopoiesis that was evident and prevented the myelosuppression brought on by the tumor’s temporal development. RJ has anti-tumor action, as the survival investigation amply shows [209]. In another study conducted by Nakaya et al. (2007), RJ was shown to inhibit the proliferation of a MCF-7 human breast cancer cell model that was induced by the environmental estrogen bisphenol A [210]. According to Wang and Chen (2019) [207], several forms of RJ extracts can considerably suppress the growth of the leukemia cell line U937. This is possible because the previously described RJ extracts stimulate the release of cytokines by mononuclear cells. HepG2 human hepatoma cells exhibit caspase-dependent apoptosis, which is induced by MRJP2 and its isoform X1 [211]. Zhang et al. (2017) [212] demonstrated that RJ treatment enhanced the antioxidative activity of many organs, including the liver and kidneys, and reduced tumor mass using a mouse breast cancer model. Furthermore, the therapy increased the activity of antioxidant enzymes, suggesting a potential connection between the antioxidant bioactive qualities and anticancer effects of RJ.

Overall, the findings related to the anticancer properties of bee products show that they are a promising source of agents with a variety of cytotoxic mechanisms. Therefore, a thorough assessment of these items against animal models of cancer is necessary in order to acquire a greater understanding of the impact of many parameters on the potencies of these natural medicines. Furthermore, as many anticancer medicines are toxic to both cancer cells and normal tissues, it is important to assess the toxicity profiles of each bee product against normal cells. The studies mentioned in this section are summarized in the following table (Table 6).

### 4.3. Bee Products as Prospective Sources of Antibacterial and Antiviral Agents

The development of bacterial and viral resistance to currently available antibiotics and antivirals is prompting the scientific community to search for effective and alternative products [213,214,215,216,217]. Bee products are among the natural items associated with these activities [218,219,220,221]. In this direction, many traditional therapeutic methods have made extensive use of bee products, including honey, propolis, bee pollen, royal jelly, beebread, and bee venom [220,222,223]. 

More than 150 distinct chemicals can be found in honey, including various forms of polyphenolic compounds, water, proteins, carbs, vitamins, and minerals [222,224]. The composition and concentration of active compounds in nectar have been found to be significantly influenced by geographic location and climate conditions [223]. As a result, honeys’ quality and, consequently, their antimicrobial and antiviral properties, can differ between one another. 

Honey has antibacterial activity against a wide range of harmful bacteria and viruses [225,226]. The biological activity of the chemical compounds found in honey, such as bacteriocins, bee defensin, methylglyoxal, 3-phenyllactic acid (PLA), and the so-called Major Royal Jelly Proteins (MRJPs), is also influenced by various physical and chemical properties, such as a low pH, a high sugar content (high osmolality), and glucose oxidase activation that produces hydrogen peroxide [227]. Honey has been shown to have remarkable antibacterial properties against Gram-positive bacteria, which are often associated with skin infections, as well as Gram-negative bacteria, including methicillin-resistant *Staphylococcus aureus* (MRSA) [228]. 

Strong antibacterial activity against *Staphylococcus aureus*, *Staphylococcus epidermidis*, *Enterobacter aerogenes*, *Salmonella enterica* serovar *Typhimurium*, *Klebsiella pneumoniae*, and *Escherichia coli* has been observed for Manuka honey, a kind of honey produced from *Leptospermum scoparium* [229]. 

Among the first studies was that of Jeddar et al. (1985) who assessed pure honey’s antibacterial properties in vitro in a pioneering investigation. The growth of bacterial colonies was investigated in a medium containing different concentrations of honey (10%, 20%, 30%, 40%, and 50% (*w*/*v*). The 40% honey content inhibited the growth of the majority of harmful microorganisms [230]. Numerous other research has been conducted in an effort to quantify and validate honey’s antibacterial properties in response to Jeddar et al.’s findings. Bogdanov (1997) examined eleven different kinds of honey’s antibacterial activity against *Staphylococcus aureus* and *Micrococcus luteus*, including typical varieties like acacia, flower, chestnut, lavender, and orange [231]. He discovered that the inhibition of the various honey variations varied from 37 to 74%. The most significant and successful component in preventing the growth of microorganisms was thought to be the pH of the honey, which ranged from 3 to 5.4.

The antibacterial efficacy of several kinds of honey made from native wild flowers cultivated in South Africa was evaluated by Basson and Grobler (2008) against *S. aureus*. The findings indicated that the osmolality and carbohydrate content of the honey required a concentration of honey over 25% in order to exhibit antibacterial action, and that the South African honey types lacked substantial bactericidal activity [232]. The antibacterial characteristics of honey are also influenced by its colloidal characteristics, (macro)molecular crowding in water pockets, and complex interactions [233,234]. Furthermore, honey, thanks to the natural characteristics of the deep eutectic solvent, increases the bioactivity of natural compounds [235,236]. 

Honey has a wide range of antibacterial actions against both Gram-positive and Gram-negative bacteria, including those that are resistant to antibiotics (MRSA). 

It has been demonstrated that honey has potent antibacterial properties in both media and cultures. Lusby et al. (2005) investigated the antibacterial activity of six types of honey variants against thirteen species of bacteria and one species of yeast [229]. Three honey varieties—Lavender, Red Stringy Bark, and Paterson’s Curse—were exposed to 15 KGY of γ-irradiation, while the remaining three—Manuka, Rewa rewa, and Medihoney—were promoted as medicinal honeys with antimicrobial properties. Different concentrations of each sample (0.1%, 1%, 5%, 10%, and 20% (*w*/*v*)) were evaluated. With *Citrobacter freundii*, *E. coli*, *Mycobacterium phlei*, and three species of *Salmonella*, there was no inhibition at the 0.1% concentration, but there was some at the 1% concentration. With the exception of *Klebsiella pneumoniae*, which oddly showed no inhibition at all, most honey samples in this investigation exhibited a gradual rise in inhibition at the highest concentration (at least 75% inhibition at a 20% concentration). 

Chestnut, Herero floral, and Rhododendron honeys from Anatolia, Turkey, were studied for their biological activity. The results showed that the honeys were active against all test microorganisms, although only a few, such *S. aureus* and *Helicobacter pylori*, showed considerable inhibition when the extracts were used [237]. Tan et al. (2009) conducted a comparative study to evaluate the antibacterial activity of Manuka honey and Malaysian Tualang honey (*Koompassia excelsa*) against a wide range of microorganisms. Their findings indicated that the MICs of Tualang honey ranged from 8.75% to 25%, suggesting that the two types of honey have similar potential for use in medical applications [238].

Several studies have been conducted to assess the effectiveness of using honey together with antimicrobial medications, with some encouraging results. When tetracycline and Manuka honey were applied together, the antibacterial potential against *S. aureus* and *Pseudomons aeruginosa* was greater than when each therapy was used alone. According to this research, a combination like this might be used as a wound healing treatment plan [239]. In a separate investigation, clinical isolates of *S. aureus*, including MRSA, showed that rifampicin resistance might be reversed when combined with subinhibitory amounts of Medihoney [240]. Additional evidence demonstrates that using honey in conjunction with antibiotics might modify antibiotic resistance. For example, Jenkins and Cooper (2012) [239] found that applying subinhibitory amounts of honey made MRSA sensitive to oxacillin.

According to reports, honey has biological effects against human pathogenic viruses, such as the most recent danger posed by SARS-CoV-2, as well as bacterial infections [241]. According to most publications, honey is a potential source of antiviral chemicals that have significant in vitro activity against the rubella virus [242] and the varicella zoster virus (VZV) [243]. 

Honey has also been shown to have antiviral efficacy against respiratory syncytial virus (RSV) [14], herpes simplex virus (HSV)-1 [244], and influenza virus [245], either when used alone or in combination with other items. Furthermore, honey may enhance the quality of life for patients with HIV by encouraging lymphocyte proliferation and preserving ideal hematological and biochemical parameters [244,246]. 

Other bee products such propolis, pollen, royal jelly, bread, and venom have all been shown to have antibacterial properties [218,221,247]. Propolis possesses antibacterial properties through two different mechanisms: either it directly interacts with specific parts of bacteria, such as by altering membrane potential and disrupting cell wall synthesis, or it indirectly acts by stimulating the host immune responses [248]. A Brazilian research team found that propolis showed antibacterial action against MRSA [249], most likely as a result of artepillin C. Propolis has been shown in independent trials by researchers in Chile and Japan to be effective against *Porphyromonas gingivalis* [250] and *Streptococcus mutans* [251], respectively. These findings raise the possibility of using propolis to treat periodontal disorders. Furthermore, it has been demonstrated that the ethanolic extract of propolis, which contains high concentrations of kaempferide, artepillin C, drupanin, and p-coumaric acid, positively correlates with its exceptional antioxidant and antimicrobial activity against a variety of pathogenic bacteria, such as *S. aureus*, *Staphylococcus saprophyticus*, *Listeria monocytogenes*, and *Enterococcus faecalis* [252]. 

Pinocembrin and apigenin are flavonoids that are present in propolis. The antibacterial activity of both of these compounds against *Streptococcus mutans* is higher than that of the polyphenols mixture or even chlorhexidine (MICs = 1.6 µg/mL), according to a study on Chilean propolis by Veloz et al. (2019) [251]. Their respective minimum inhibitory concentrations (MICs) were 1.4 µg/mL and 1.3 µg/mL. A number of investigations have shown that isolated pinocembrin has antibacterial action against *S. aureus*, *S. mutans*, *Streptococcus sobrinus*, *Enterococcus faecalis*, *Listeria monocytogenes*, *Pseudomonas aeruginosa*, and *K. pneumonia* [253,254,255,256]. *Salmonella enterica* serotype *Typhimurium*, *P. aeruginosa*, *K. pneumoniae*, *Proteus mirabilis*, and *Enterobacter aerogenes* are among the Gram-negative bacteria that are inhibited by isolated apigenin [257].

Propolis has high concentrations of esters and cinnamic acid. Numerous investigations have demonstrated the antimicrobial properties of cinnamic acid against a variety of pathogens, including *Streptococcus pyogenes*, *Bacillus* spp., *Mycobacterium tuberculosis*, *Aeromonas* spp., *Vibrio* spp., *E. coli*, *L. monocytogenes*, *Micrococcus flavus*, *P. aeruginosa*, *S. enterica* serotype *Typhimurium*, *Enterobacter cloacae*, and *Yersinia ruckeri* [258,259,260]. By disrupting the bacterial cell membrane, cinnamic acid and its derivatives prevent ATPases, cell division, and the production of biofilms. They further exhibited anti-quorum sensing activity [261].

Propolis has been shown to have antiviral activity against a wide range of human pathogenic viruses. These viruses include human herpesviruses [13], influenza viruses [14,15], HIV [17], human T-cell leukemia-lymphoma virus type 1 (HLTV-1) [18], Newcastle disease virus (NDV) [19], RSV [20], poliovirus (PV)–type 1 [21], and dengue virus (DENV) [22]. Flavonoids found in propolis and honey, including quercetin, rutin, and naringin, have recently been proposed as possible adjuvant treatments for SARS-CoV-2 [262]. 

Published research indicates that bread and bee pollen have potent antibacterial properties against a range of bacterial and viral disease agents [218]. Like propolis and honey, bee pollen and bread have varying antibacterial properties that are mostly influenced by the location from which the samples were gathered as well as the solvents used during the extraction procedure [218]. Commercial Spanish and Portuguese bee pollen has been shown to have some properties against pathogens, such as *Candida glabrata* and *Staphylococcus aureus* [263]. Fatrcová-Šramková et al. (2013) found that monofloral bee pollen had an antibacterial effect against harmful microorganisms [264]. For instance, the most sensitive bacteria were *Staphylococcus aureus* when exposed to an ethanol extract (70%) of poppy pollen (Papaver, Papaveraceae), and *Salmonella enterica* when exposed to methanol extracts (70%) of rape bee pollen (*Brassica napus*, Brassicaceae), and 70% of sunflower pollen (*Helianthus annus*, Asteraceae). 

Bee pollen extracts had less of an effect on *Escherichia coli*, *Pseudomonas aeruginosa*, and *Listeria monocytogenes*. An 80% ethanol extract of bee pollen was shown to have antibacterial properties against *Pseudomonas aeruginosa*, *Bacillus subtilis*, *Staphylococcus aureus*, and *Klebsiella* sp. [188]. But when tested at concentrations ranging from 0.02% to 2.5% (v/v), pollen was found to have no antimicrobial effects on bacteria (*Escherichia coli*, *Bacillus cereus*, *Bacillus subtilis*, *Salmonella typhimurium*, *Staphylococcus aureus*, *Yersinia enterocolitica*, *Enterococcus faecalis*, and *Listeria monocytogenes*) or fungi (*Saccharomyces cerevisiae* and *Candida rugosa*, *Aspergillus niger*, and *Rhizopus oryzae*) [265]. As a result, it is clear that bee pollen’s antibacterial action depends on concentration. 

The ability of pollen to inhibit bacteria is most likely linked to the enzyme glucose oxidase, which is generated by honeybees. When the pollen granules develop, it is added to the pollen [266]. Additionally, it has been shown that flavonoids and phenolic acids are linked to microbiological activity [265]. Degradation of the cytoplasm membrane, which results in potassium ion loss and the start of cell autolysis, is the mechanism by which flavonoids and phenols work against bacterial and fungal cells.

The growth of *Streptococcus pyogenes* I.S.P. 364-00 was decreased by Chilean bee pollen extracts, but no biological activity was shown against *S. aureus* ATCC 25923, *P. aeruginosa* ATCC 27853, or *E. coli* ATCC 25922 [267]. Remarkably, a clinical strain of *E. coli* CCM 3988 was susceptible to the excellent antibacterial properties of the Slovakian bee pollen extract that was used [268]. 

However, a common finding across several investigations is that, with a few notable exceptions [264,269], bee pollen has a far stronger antibacterial effect on Gram-positive bacteria than it does on Gram-negative ones [263,267,270,271]. It is crucial to remember that almost all of the antibacterial data were produced in vitro, thus it is essential to use the in vivo vertebrate [272,273,274,275] or invertebrate [276,277,278,279,280,281] model systems that are now available to establish the antibacterial activity of bee products. Bee pollen and beebread have been shown to have antiviral properties in addition to their antibacterial ones. Bee pollen extracts from Korean Papaver rhoeas were shown to be somewhat efficient against influenza viruses (strains of H1N1, H3N2, and H5N1), whereas bee pollen from the date palm was found to be active against HSV-1 and HSV-2 [282,283]. 

The antiviral properties of bee pollen were probably attributed to flavonoids such as quercetin, kaempferol, galangin, and luteolin. Luteolin is a potential anti-influenza candidate since it has been shown to be one of the most effective neuraminidase inhibitors of the influenza virus [283]. Furthermore, it has been shown that quercetin interacts with the HA2 component of hemagglutinin to prevent the influenza virus from entering host cells [284]. The suppression of hemagglutinin, which is mediated by quercetin, may be a key factor in preventing the interaction between hemagglutinin and sialic acid, which is necessary for influenza virus’ entrance into the body. Such a mechanism will be important in the future pharmacological treatment of influenza virus infections due to the rising incidence of viral resistance to the anti-influenza medications now on the market. 

Bee venom also exhibits antibacterial and antiviral properties. The main component of bee venom that has antibacterial effects is the peptide melittin [285]. The mechanism of melittin’s antibacterial action is due to its ability to damage cellular membranes. Gram-positive bacteria are more susceptible to melittin than Gram-negative bacteria because of the composition of their cell membrane. Melittin has a higher affinity for the peptidoglycan layer found on the membrane of Gram-positive cells than for the lipopolysaccharide layer found on the membrane of Gram-negative bacteria. Melittin and bee venom have been shown to be successful in eliminating 86% of Gram-positive bacteria and 46% of Gram-negative bacteria, as reported by Fennell et al. (1968) [286]. For Gram-positive bacteria, one milligram of melittin has the same antibacterial effect as 0.1 to 93 units of penicillin. The proline residue at position 14 has been shown to be essential to melittin’s antibacterial activity. When compared to the natural peptide, its absence from a melittin derivative dramatically decreased the derivative’s anti-microbial efficacy.

Research has been carried out on the antibacterial efficacy of melittin against a variety of bacteria, such as *Escherichia coli*, *S. aureus*, and *Borrelia burgdorferi* [247,287,288]. Melittin demonstrates antibacterial activity against methicillin-sensitive *S. aureus* (MSSA), MRSA, and *Enterococcus* spp. bacteria with MICs of 0.5–4, 0.5–4, and 1–8 g/mL, respectively [289]. In Table 7, the hive products and their components that have been assayed and mentioned in this section are presented.

### 4.4. Antiparasitic Potential of Bee Products

In the nations with subtropical, tropical, and temperate climates, parasitic infections continue to rank among the most difficult public health problems [290,291,292,293,294,295]. The pharmacological and chemical properties of bee-related products, a prospective source of naturally occurring bioactive compounds, have attracted a lot of attention in recent decades as potential alternative antiparasitic treatments [296]. In numerous cultures all over the globe, bee-related items have long been utilized as herbal treatments to cure various infectious ailments since ancient times [297]. Propolis, bee venom, bee pollen, and honey have undergone in-depth research to determine their antiparasitic properties against protozoa and helminths, the two most prevalent types of parasites that infect people. Numerous studies have demonstrated the scientific efficacy of bee products in the treatment of a wide range of infectious diseases, including amebiasis, giardiasis, cryptosporidiosis, echinococcosis, leishmaniasis, toxocariasis, plasmodiasis, toxoplasmosis, blastocystis infection, and schistosomiasis (chagas disease) [298,299,300,301]. 

The chemical components of bee products have been directly linked to their therapeutic abilities. Regional differences in the antiparasitic effects of bee products, however, suggest that the chemical components of bee products are complex and vary depending on their botanical source and geographical origins [302,303,304]. The vegetation surrounding the beehive, collection time, soil diversity, geoclimatic conditions or seasons in the collection area, the species of bees, and specific flora present at the harvesting location are other factors that have been reported to influence the dissimilarity of the physicochemical characteristics of the bee products [305,306,307]. The kind and source of the parasites used in the studies, as well as the technique used for preparation, also have a major impact on variations in the concentration of beneficial bee products [308,309]. To obtain propolis extracts, for instance, a variety of extraction techniques are used, ranging from traditional separation techniques using organic solvents like ethanol to more advanced techniques like the supercritical fluid extraction approach [310]. The quantity of active ingredients in the extract may be affected by the extraction techniques, which might alter the extracts’ biological activity [310]. Finally, the biological qualities’ magnitude is also determined by the sort of bee products involved. According to some studies, the chemical composition of the many Brazilian propolis varieties—such as red, green, and brown—varies, and as a result, so does their effectiveness against human parasites [311,312]. The flavonoid and phenolic components of bee products are thought to assist a number of suggested pharmacological mechanisms by which they combat protozoan infections [313,314,315]: The parasite is killed by (1) activating the macrophages that produce ROS (especially superoxide dismutase) and nitrogen metabolites [316]; (2) the alteration of angiogenesis in the affected tissue [316]; (3) stimulation of immunomodulatory effects, which affect the production of interferon-, tumor necrosis factor-, IL-1, IL-4, and IL-17 [317,318]; (4) induction of mechanisms in parasites that resemble apoptosis [316]; and (5) disruption of membrane in parasites [319].

For the treatment of some organisms such as *Trypanosoma* and *Plasmodium falciparum*, the enzyme phospholipase A2 found in bee venom can be utilized as an anti-parasitic treatment [320,321]. It is also important to draw attention to research on the use of bee venom against *Toxoplasma gondii*. Live tachyzoites are harmed by bee venom, as demonstrated by Hegazi et al. (2014) [322,323]. Table 8 summarizes the studies cited in this section.

### 4.5. Antioxidant Properties of Bee Products

Propolis’s antioxidant qualities have been thoroughly examined and demonstrated using the DPPH, ABTS+, FRAP, and ORAC techniques [49,325,326,327]. Propolis extracts were shown to have an antioxidant capacity comparable to ascorbic acid or butylated hydroxytoluene, two synthetic antioxidants, in the same in vitro tests [49,328]. Propolis’s content determines its significant antioxidant potential, yet research attempting to identify the unique correlations between these two factors has not produced consistent results [325,329]. The total phenolic content of propolis extracts ranged from approximately 30 to 200 mg of gallic acid equivalents (GAE)/g of dry weight, and the flavonoid content from approximately 30 to 70 mg of quercetin equivalents (QE)/g, according to data from the literature. The range of DPPH free radical-scavenging activity was found to be between 20 and 190 μg/mL [49,325,326,330]. The robust antioxidant activity of Brazilian green propolis appears to be attributed to 3,4,5-tricaffeoylquinic acid, 3,5-dicaffeoylquinic acid, 4,5-dicaffeoylquinic acid, and artepillin C, according to Zhang et al. (2017) [329]. In contrast to Brazilian propolis, the total polyphenol and total flavonoid concentrations of poplar propolis seem to have a significant impact on its antioxidant activity [3,34,38]. According to the findings of Fabris et al. (2013) [331], propolis samples from Europe (Italy and Russia) had comparable polyphenolic compositions and, therefore, comparable levels of antioxidant activity, but propolis samples from Brazil had lower levels of polyphenolic content and, hence, weaker antioxidant qualities. The standardization of propolis’s appears to be a major issue overall because it depends so heavily on a variety of variables, including bee species, plant origin, geographic location, temperature variation, seasonality, and storage conditions [49,325,326,330,332,333]. Using samples of Brazilian propolis obtained from the same location, Bonamigo et al. (2017) [49,326] investigated the antioxidant activity of the ethanol extract based on the species of bees, *Scaptotrigona depilis*, *Melipona quadrifasciata anthidioides*, *Plebeiadroryana*, and *Apis mellifera*. It was discovered that the investigated samples varied in terms of their composition, free radical-scavenging activity, and capacity to prevent lipid peroxidation. Propolis derived from *A. mellifera* often had the highest activity. In response, Calegari et al. (2017) [333] discovered that Brazilian propolis samples produced in both March and April had a different color to those produced in May and June, as well as a higher content of total phenolic compounds and antioxidant capacity. This finding suggested that the month of production had an impact on the chemical composition of propolis, an effect that can be explained by temperature variations. The researchers also found that, compared to colonies that did not receive food supplementation, those that did so every three days throughout the year showed noticeably greater levels of total phenolic and flavonoid content as well as antioxidant capacity [333]. Furthermore, the kind of solvents utilized for the extraction has a significant impact on the chemical makeup and biological characteristics of propolis extracts [325,330,334]. Aqueous ethanol is the most often used solvent for extracting propolis, especially when the concentration is between 70 and 75%. Other solvents that are sometimes employed include ethyl ether, water, methanol, hexane, and chloroform. According to Sun et al. (2015) [325], Beijing propolis extraction yields (the weight ratio of the dry extract to the weight of the raw extract) ranged from 1.8% to 51% and showed a tendency to rise as the ethanol concentration increased. Total flavonoid and polyphenol content varied significantly, ranging from 4.07 to 282.83 mg of rutin equivalents (RE)/g and 6.68 to 164.20 mg of GAE/g, respectively. The highest concentration was found in 75% ethanol solvents; it was slightly lower in 95% and 100% ethanol solvents, and lowest in water solution. According to tests using DPPH, ABTS, FRAP, oxygen radical absorbance capacity (ORAC), and cell antioxidant activity (CAA), the 75% extract exhibited the best antioxidant capacity. The “polar paradox” arises from the fact that polar antioxidants perform better in nonpolar matrices while nonpolar antioxidants perform better in polar matrices [335].

Nevertheless, significant variations were noted, even when the same solvent or one with a similar polarity was used to extract various propolis sample types [330], suggesting that additional factors may also have an impact in addition to the solvents’ molecular structure. For instance, Bittencourt et al. (2015) [330] showed that while partitioning with hexane dramatically reduced the quantity of antioxidant components in green propolis extract, partitioning with dichloromethane improved the extraction of antioxidant chemicals, particularly in brown propolis. 

Bee pollen also proves to be extremely important due to its antioxidant characteristics. The antioxidant properties of bee pollen, which include the inactivation of oxygen radicals, may be attributed to the activity of antioxidant enzymes and the presence of secondary plant metabolites such carotenoids, phenolic compounds, vitamin C, vitamin E, and glutathione [188]. The most prevalent and well researched class of low molecular weight polyphenols is flavonoids. 

Quercetin, pinocembrin, apigenin, chrysin, galangin, kaempferol, isorhamnetin, caffeic acid, and caffeic acid phenethyl ester (CAPE) are a few of the compounds found in bee pollen [191,336]. Research has shown that the flavonoids found in bee pollen have the ability to deactivate electrophiles and scavenge reactive oxygen species (ROS), preventing them from developing into mutagens [263]. By capturing the free radical chain oxidation, the hydrogen from the phenolic hydroxyl groups of flavonoids forms stable end products that prevent more oxidation. Additionally, flavonoids bond to metal ions, potentially eliminating harmful metals from the body [336]. Flavonoids operate as a protective factor against carcinogens and genotoxic chemicals while also bolstering the body’s defense against free radicals [115,337]. Duclos et al. (2007) observed reduced oxidative stress and increased antioxidation in prostatic secretions and semen after the administration of bee pollen extracts in an experimental clinical trial [338]. In bee pollen extracts, ethanol, and methanol/water, the flavonoid concentration rises noticeably [339]. As a result, pollen extracts have more antiradical activity than pollen that has been gathered by bees. Nonetheless, the antioxidant impact of pollen from bees varies greatly across different species [340]. The antioxidant effect in the research by Fatrcová-Šramková et al. (2013) declined in the following order: >*Papaver somniferum* (Papaveraceae) > *Helianthus annuus* (Asteraceae) > *Brassica napus* subsp. *napus* (Brassicaceae) [264]. Flavonoids’ ability to reduce inflammation might be attributed to quercetin’s impact, since this compound is known to block the metabolism of arachidonic acid [111]. The anti-inflammatory effect is caused by a reduction in the amount of arachidonic acid, which also lowers the level of prostaglandins that promote inflammation [115]. Thus, after applying bee pollen, positive results are seen in terms of a reduction in local pain and the inhibition of platelet aggregation [341,342]. 

Numerous in vitro investigations have examined the antioxidant capabilities of bee pollen utilizing DPPH, ABTS+, and FRAP techniques. It is commonly recognized that the content of bee pollen affects its antioxidant activity. However, a significant variance in the outcomes of the numerous studies that have been conducted to ascertain the composition and characteristics of various bee pollen samples has been demonstrated. While some research [343,344,345] revealed no significant associations, other studies [346] identified a high positive link between the overall concentration of phenolic compounds and the antioxidant capacity of bee pollen. It was discovered that phenylpropanoid concentration was connected with the overall antioxidant activity as determined by the prevention of linoleic acid peroxidation [347]. According to Sousa et al. (2015), flavonols may function as prooxidants in their oxidized and reduced forms, respectively, whereas anthocyanins function as antioxidants [348]. It has also been discovered that the characteristics and content of bee pollen are influenced by the type of plant that provides it, as well as the environmental factors that the plants grow in, such as the soil and climate [349]. Another aspect influencing these qualities is the time of harvest [68,263,343,344,350]. There have also been reports on the possible impact of freezing and/or freezing followed by dehydration on the composition and characteristics of bee pollen. Freezing or freezing and then drying had no effect on the chemical composition, although freezing and further drying increased the antioxidant activity. The observed benefits were ascribed by the researchers to a decrease in moisture, which in turn led to a concentration of antioxidants [346]. Significant dispersion was also observed in the variations in polyphenolic compounds, both overall and among specific types. For instance, LeBlanc et al. (2009) [351] reported that pollen from mimosa bees contained 34.85 mg/g of polyphenolics expressed as gallic acid equivalents, whereas only 19.48 mg/g and 15.91 mg/g were found in pollen from yucca and palm bees, respectively. Pollen from *Pyrus communis* bees had a flavonoid level of 1349 mg/100 g, but pollen from *Lamium purpureum* bees only had a flavonoid value of 171 g/100 g [347]. Depending on the time of harvesting and the main pollen type, distinct samples of bee pollen gathered in northeastern Brazil over a nine-month period (January–November) were shown to have varying flavonoid profiles [352].

As with propolis, studies have revealed that the kind of extraction solvent employed can have a significant impact on the characteristics of the pollen extract. This is related to the various solubilities of certain bee pollen components in polar solvents. It has been demonstrated that using nonpolar solvents produced extracts with extremely poor antioxidant activity, whereas using polar solvents allowed for the extraction of extracts with higher antioxidant characteristics. Even when solvents with identical polarity were used, significant variations were still noted [353,354,355]. Kim et al. (2015) carried out research on the ideal conditions for bee pollen extraction [356]. In their experiment, n-hexane, dichloromethane, ethyl acetate, and n-butanol were the solvents of differing polarity that were used to progressively partition the entire extract produced by extraction with 80% methanol (twice). The fractions with the highest activity were ethyl acetate and n-butanol. Therefore, the best extraction conditions were determined using the response surface approach and the Box–Behnken design (BBD) with three level-three factors. Temperature, time, and the amount of ethyl acetate in the methanol were the factors. The solvent concentration turned out to have the biggest influence, and the following settings were found to be ideal: an ethyl acetate content of 69.6% in methanol at 10.0 °C for 24.2 h. The extract prepared under the theoretically estimated circumstances demonstrated antioxidant activity and tyrosinase inhibition that was very similar to those expected by statistical approaches, confirming the calculated values experimentally [356]. The animal studies also demonstrated that the characteristics of the extract depended on the extraction solvent utilized. Bee pollen that was mass administered orally to rats with induced hind paw edema showed modest suppressive effects; the water extract had almost no impact; and the ethanol extract was most effective [357]. 

Among the information at hand, a few papers [358,359,360] attest to the function of royal jelly in scavenging free radicals. Liu et al. (2008) [361], for example, looked at the antioxidant qualities of royal jelly as well as its ability to scavenge radicals such as hydroxyl, superoxide, and DPPH radicals. Its reducing capacity, prevention of linoleic acid oxidation, and superoxide dismutase activity were also assessed by the researchers. The acquired findings were compared based on the period of harvest (24, 48, and 72 h) following the larval transfer from the queen cell cups to the bee hives and the larval age (1, 2, or 3 days). The authors observed that there was an inhibitory impact on the generation of superoxide radicals (ranging from 23.9 to 37.4%) and hydroxyl radicals (48–68%), in addition to a DPPH radical-scavenging effect (in the range of 43.0–62.8%). Additionally, it was shown that the royal jelly sample inhibited the peroxidation of linoleic acid (8.6–27.9%). In every instance, samples collected from the youngest larvae, one-day old, that were kept in bee hives for the shortest amount of time, 24 h, showed the highest scavenging effect of RJ. Furthermore, it was found that the royal jelly samples exhibited the highest decreasing power. Conversely, the superoxide dismutase (SOD) activity of the royal jelly that was extracted 72 h after the 3-day-old larvae were transferred was noticeably higher than that of the other samples. As a result, the scientists hypothesized that antioxidant substances other than SOD may be responsible for the royal jelly’s ability to scavenge superoxide radicals. 

Guo et al. (2009) [359] discovered that peptides produced by employing protease N to hydrolyze royal jelly proteins have potent antioxidant qualities. The discovered peptides’ antioxidative qualities were investigated in terms of processes including metal-chelating activity and the ability to scavenge radicals like hydroxyl, superoxide, and hydrogen peroxide. Significant hydroxyl radical-scavenging activity was demonstrated by twelve of the derived peptides, and significant hydrogen peroxide-scavenging activity was demonstrated by the three dipeptides (Lys-Tyr, Arg-Tyr, and Tyr-Tyr) that have Tyr residues at their C-termini. Nevertheless, no appreciable superoxide anion radical scavenging or metal chelating properties of the isolated peptides were seen in this investigation. The scientists came to the conclusion that di- and tripeptides could be more antioxidatively active than the amino acids that make them up. 

Strong antioxidant activity (AOA) compounds have also been found in bee venom [78]. This reaction is caused by melittin, apamin, and phospholipase A2. The antioxidant action is supported by many mechanisms, such as hydrogen donation, metal ion chelation, single oxygen quenching, free-radical scavenging, and serving as a substrate for superoxide and hydroxyl radicals. The compounds’ capacity to suppress lipid peroxidation (a process involving chemical species with an independent existence and one or more unpaired electrons or an odd number of electrons, known as free radicals) and to increase the activity of the important enzyme superoxide dismutase (which lowers radical damage by eliminating the superoxide radical in practically all cells exposed to oxygen) could underlie the antioxidant effect. But in addition to these, bee venom also includes other compounds that serve as antioxidants. As an illustration of its antioxidant activity, vitellogenin gives mammalian cells a defense against reactive oxygen molecules by directly shielding them from oxidative stress.

The AOA of bee venom has only been measured using standard assays in three recent studies [362,363,364]. All of the samples had antioxidant characteristics, which did not seem to be connected to any of the particular elements that were found and analyzed in the same samples. The evidence suggests that melittin exhibits a very low AOA when compared to extracts of bee venom; this might be due to the involvement of other venom components as well [78]. Therefore, different results among bee venom samples may be caused by additional tiny molecules that are also implicated in the reported bioactivities, along with synergistic or antagonistic effects at particular doses. Rekka et al.’s (1990) study is one of the earliest ones that was carried out [365]. Rekka and colleagues have shown that the venom of honeybees significantly inhibits the process of nonenzymatic lipid peroxidation. It also has high hydroxyl radical scavenging capabilities, as evidenced by its competition for HO (hydroxyl radicals) with dimethyl sulfoxide. The hypothesis that antioxidant activity contributes to the anti-inflammatory characteristics of honey bee venom—which is mostly recognized for its capacity to reduce interleukin-1 production in vitro—may find additional support in our findings [365]. 

Antioxidant activity has been examined in combination with other criteria in various research. For instance, for a total of 20 weeks, El-Hanoun et al. (2020) [366] subcutaneously delivered 0.1, 0.2, and 0.3 mg per rabbit twice a week. Throughout the experiment, many antioxidant activity markers were assessed, including total antioxidant capacity (TAC), glutathione S-transferase (GST), glutathione content (GSH), glutathione peroxidase (GPx), superoxide dismutase (SOD), malondialdehyde (MDA), and thiobarbituric acid reactive substances (TBARS). The results showed that the treated rabbits had higher levels of GST and GSH. The readings of TBARS and MDA were also lower. The antioxidant qualities of BV were validated by these results.

## 5. Common Applications of Hive Products

Research has shown that honey has healing qualities for wounds [367]. According to several studies, honey’s ability to promote healing is attributed to its antibacterial activity and ability to keep wounds moist [225]. Several research findings thus support the antibacterial characteristics of honey. Hydrogen peroxide is produced enzymatically in honey [368]. The low pH level of honey and its high sugar content are sufficient to inhibit the development of microorganisms [225].

Honey, because of its high viscosity, occupies a significant position among other hive products as a conventional protective barrier against infection. Its immunomodulatory characteristics are likewise important in supporting this activity [369].

Because it is easier to apply, more pleasant than other dressings, antibacterial, self-sterile, nourishing, and non-irritating, honey is a good wound therapy. Honey therapy has been found to be effective for almost all types of wounds, including burns, amputation, bed sores/decubitus ulcers, burns, chill blains, burst abdominal wounds, cracked nipples, fistulas, diabetic malignant, leprosy, traumatic, cervical, varicose, and sickle cell ulcers, septic wounds, surgical wounds, and wounds of the abdomen wall and perineum [370]. When honey is applied to wounds, it promotes healthy tissue granulation and, after 7 to 10 days, the wounds become bacteriologically sterile. The antibiotic gentamicin was shown to be less efficient than honey as an antibacterial agent against a number of species of *Pseudomonas* and *Staphylococcus* [371].

On the other hand, an extensive immersion of wounds and abscess cavities in honey, sometimes combined with castor oil to aid in dressing, was discovered to have the following benefits: first, it prevented the cross-infection of wounds frequently encountered with conventional therapy, as honey forms a mechanical and/or chemical barrier to infectious agents (effective in initiating the healing process in non-healing ulcers, leprosy, and diabetic patients); second, it resulted in a shorter treatment duration and consequently decreased hospitalization [372]. 

Research evidence has also demonstrated the possible application of honey in gastroenterological medicine. *Helicobacter pylori* infection can cause problems such as gastritis, duodenal ulcers, and stomach ulcers. These days, the Gram-negative, microaerophilic bacteria *Helicobacter pylori* (*H. pylori*) colonize more than half of the human population. Infections are often lifelong and cause chronic active illness if left untreated. While the majority of persons infected with these bacteria do not show any symptoms, 5–10% of those infected go on to develop serious gastro-duodenal disorders, such as gastric and duodenal ulcers, and gastric lymphomas [373]. Due to the prevalence of antibiotic resistance, conventional treatments for the eradication of *H. pylori*, such as triple therapy of proton pump inhibitors like omeprazole and two antibiotics, clarithromycin and either amoxicillin, are far from satisfactory; hence, there is a search for alternative treatments. Remedies made from bee honey have the potential to yield novel chemicals that might be helpful in treating *H. pylori* infections [44].

Research on animals and in clinical settings has demonstrated that honey lowers stomach acid output. Additionally, using honey as a dietary supplement has been shown to be effective in treating stomach ulcers. It has been reported that 600 patients with stomach ulcers treated with oral in honey had up to an 80% healing rate [374].

The restorative properties of pollen and those it shows against prostatic disorders are also well known. Pollen is useful in treating prostate issues ranging from infections and swelling to cancer, according to several decades’ worth of observations in Western European nations and a few scientific trials. It has been demonstrated that the phytochemicals in bee pollen, such as lycopene, beta-sitosterol, and other phytosterols, as well as a variety of flavonoids, limit the growth of prostate tissue and lower discomfort, inflammation, and the risk of prostate cancer [375].

Antioxidant compounds abound in pollen. Antioxidants have the ability to offer defense against this oxidative damage, sometimes offering substantial defense. Pollen is a promising treatment option for age-related ailments including chronic fatigue and arteriosclerosis because of its anabolic, growth-stimulating qualities and cardiovascular health benefits. As an interesting side note, bee pollen has long been used as an anti-aging meal since it seems to have strong antioxidant properties [376]. 

Bee venom is mostly used to treat a variety of inflammatory conditions, including arthritis, bursitis, tendinitis, rheumatoid arthritis, dissolving scar tissue, multiple sclerosis, and osteoarthritis, because of its anticoagulant and anti-inflammatory qualities [377]. Additionally, studies using a variety of animal experimental models with inflammatory disorders showed that administering bee venom was a successful way to decrease inflammation (arthritis). 

Propolis is often used in products for the treatment of oral conditions and in products used in common hygiene regimes. Oral care solutions containing propolis are commercially available globally and are marketed under a variety of brand names [174].

Hive products find wide use as ingredients in cosmetic formulations. One of the main ingredients in skin care products is honey, a bee product with a high nutritional value and regenerative qualities. Fruit acids, trace elements, and a high carbohydrate content are what give it its nutritious and restorative properties. Osmosis stimulates the microcirculation of the cutaneous tissue, improving its oxygenation and feeding. This also stimulates metabolic activities, which increase regeneration processes and contribute to the removal of toxic metabolites. Honey also contains hygroscopic qualities that allow it to absorb metabolites and detoxify the skin tissue. When this happens, the skin becomes more tense, more elastic, more vibrant, and creases are smoothed out [378]. In addition to its antibacterial and anti-inflammatory properties, propolis also promotes cicatrization and lessens discomfort. Propolis enhances the biosynthesis of glycosaminoglycans, which are necessary for tissue development, granulation, and wound healing. Moreover, propolis causes type I and type III collagen to accumulate more in the lesion matrix, which can speed up the healing process. Collagen I and III are two crucial components of skin connective tissue that are necessary for keratinocyte migration and re-epithelialization. Reepithelization may be aided by the topical apitherapeutic use of propolis ointment for burn treatment [379]. Additionally, propolis works wonders for treating acne vulgaris. After applying an ethanol extract of propolis to the skin, researchers verified the suppression of *Cutibacterium acnes*, a bacterium that is essential to the development of acne vulgaris [380]. 

## 6. Future Perspectives

Traditional medical practices have made extensive use of bee products, including honey, propolis, bee pollen, royal jelly, beebread, beeswax, and bee venom. Over the past century, bee products have witnessed an increase in interest due to their possible medicinal and pharmacological benefits. In order to improve the use of bee products in disease management, the primary active component or components responsible for the anticancer, antibacterial, antiviral, and antiparasitic properties of bee products need to be clearly and uniformly elucidated. This is made possible through improvements to research tools and our growing understanding of biological processes. The usage of bee products has been hampered by two fundamental issues: the standardization issue in the food, cosmetic, and pharmaceutical industries and the bioavailability of certain hive products. The same hive product may have a different composition in relation to geographic origin and the environmental factors that can strongly affect the chemical composition. Concerning bioavailability, if fermentation can help to increase the availability of components in pollen, nanotechnological approaches that lead to the creation of a nanoformulation should be considered for propolis [381,382,383]. In addition, research must be carried out to establish the best way to employ bee products to treat infections and cancer, as well as the ideal dosage for them. This is important information to help translate the bench-to-bedside trial outcomes to rea-world applications. Other factors limiting their use are the dearth of in vivo studies. Filling all these gaps would certainly help to make the use and medical “prescription” of these natural remedies more customary.

## 7. Conclusions

Products from the hive prove to be extremely interesting in terms of composition and related nutraceutical and pharmacological properties. The evidence from the literature points to the possibility of a plethora of applications for their medical use. Further research to implement apitherapy and make it more common must, therefore, be encouraged because it could lead to particularly successful and satisfying outcomes and mitigate important problems such as those related to drug resistance.

## Figures and Tables

**Figure 1 pharmaceuticals-17-00646-f001:**
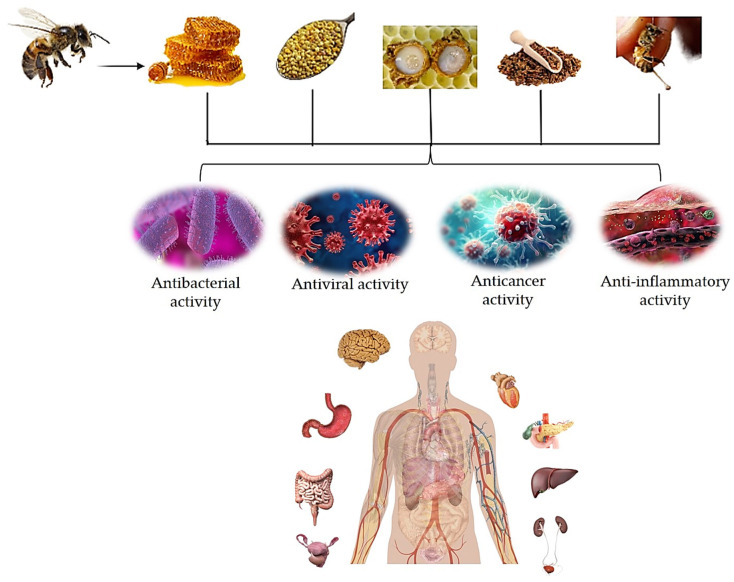
Some general actions of hive products and anatomical sites where they exert their effects.

**Table 1 pharmaceuticals-17-00646-t001:** Principal honey components.

Class	Substances	References
Carbohydrates	glucose, fructose, sucrose, gentiobiose, isomaltose, kojibiose, laminaribiose, maltose, maltulose, nigerose, and trehalose, and trisaccharides, such as centose, erlose, isomaltosylglucose, isopanose, 1-ketose, maltotriose, panose, and theanderose	[36]
Peptides	defensin-1 and royal jelly protein (MRJP)	[37]
Enzymes	acid phosphatase, amylase, catalase, diastase, glucose oxidase, invertase, and sucrose diastase	[38]
Phenolic acids	caffeic, cinnamic, ferulic	[39]
Amino acids	alanine, asparagine, glutamine, glycine, and proline	[40]
Organic acids	citric, gluconic, acetic and formic acid	[30,41,42]
Flavonoids	myricetin, naringenin, apigenin, hesperetin, galangin, luteolin, quercetin, and kaempferol	[30,43]
Minerals	P, Na, Ca, K, S, Mg, Cl, Si, Rb, V, Zr, Li, and Sr	[44,45]
Vitamins	mostly B-group (B1, B2, B3, B5, B7, B6, B8, B9)	[46]
Volatile substances	alcohols, aldehydes, benzene and its derivatives, terpene and its derivatives, ketones, pyran, furan, and acid esters	[47]

**Table 2 pharmaceuticals-17-00646-t002:** Principal propolis components.

Class	Substances	References
Enzymes	acid phosphatase, adenosine triphosphatase, glucose-6-phosphatase, and succinic dehy-drogenase	[8]
Fatty acids	arachidonic, cis-13, 16 docosadienoic, cis-11,14,17-eicosatrienoic, cis-5,8,11,14,17-eicosapentaenoic, eicosadienoic, elaidic, heneicosylic, linoleic, oleic, pal-mitic, and palmitoleic acid	[8,53]
Phenolic acids, and flavonoids	caffeic acid, chlorogenic acid, cinnamic acid, gallic acid, 4-hydroxybenzoic acid, 4-hydroxyhydrocinnamic acid, 4-hydroxybenzoic acid-methyl ester, apigenin, chrysin, acacetin, catechin, daidzein, formononetin, naringenin, galangin, neoflavonoid, neovestitol, macarangin, kaempferol, luteolin, liquiritigenin, myricetin, pinocembrin, rutin, and quercetin	[8,54,55,56,57]
Terpenoids	trans-ꞵ-terpineol, linalool, camphor, junipene, γ-elemene, α-ylangene, valencene, 8-β*H*-Cedran-8-ol, 4-βH,5α-eremophil-1(10)-ene, α-bisabolol, α-eudesmol, α-cadinol, patchoulene, manoyl oxide, ferruginol, ferruginolone, 2-hydroxyferruginol, 6/7-hydroxyferruginol, sempervirol, abietic acid, 18-succinyloxyabietadiene, 18-succinyloxyhydroxyabietatriene, 18-hydroxyabieta-8,11,13-triene, imbricataloic acid, diterpenic acid, neoabietic acid, labda-8(17),12,13-triene, hydroxydehydroabietic acid, dihydroxyabieta-8,11,13-triene, 13(14)-dehydrojunicedric acid, dehydroabietic acid, lupeol, lupeol acetate, lanosterol, germanicol acetate, germanicol, β-amyrin acetate	[53,58,59,60,61]
Minerals	Mg, Cu, F, Ca, K, Na, Mn, and Zn	[8,62]
Vitamins	B1, B2, B6, C, and E	[8,63]

**Table 3 pharmaceuticals-17-00646-t003:** Principal pollen components.

Class	Substances	References
Carbohydrates	raffinose, erlose, isomaltose, maltose, melibiose, melezitose, rhamnose, ribose, trehalose, and turanose	[65,66,70]
Amino acids	valine, phenylalanine, tryptophan, histidine, isoleucine, leucine, lysine, and methionine	[65,71]
Lipids	Myristic, palmitic, stearic, oleic, linolenic, linoleic acids, arachidonic, behenic, capric, caproic, caprylic, 11-eicosenoic, eicosatrienoic, elaidic, lauric, and lignoceric	[65,72]
Flavonoids	isorhamnetin, kaempferol, quercetin, luteolin, naringenin, apigenin, epicatechin, hesperetin, and catechin	[77]
Phenolic acids	rosmarinic, syringic, caffeic, chlorogenic, ferulic, gallic, p-coumaric, p-hydroxybenzoic, protocatechuic, and vanillic acid	[65,77]
Vitamins	vitamin B1, B2, B3, B5, B7, B6, B8, B9, C, D, E	[65,75]
Minearls	Fe, Cu, Cr, Mn, Se, Si, Zn, Ca, K, Mg, Na, P	[65,76]

**Table 4 pharmaceuticals-17-00646-t004:** Principal royal jelly components.

Class	Substances	References
Carbohydrates	Fructose, glucose, erlose, gentobiose, isomaltose, maltose, melezitose, raffinose, and trehalose	[82]
Proteins	MRJPs 1–9	[35]
Peptides	apisimin, jelleines, and royalisin	[35]
Fatty acids	trans-2-decenoic acid, 24-methylenecholesterol, 10-hydroxy-trans-2-decenoic acid (10-HDA) and 10-hydroxydecanoic acid (HDAA), and sebacic acid, 3-hydroxydecanoic, 9-hydroxy-2-decenoic, 8-hydroxyoctanoic, and 9-hydroxydecanoic acid	[87]
Flavanones, flavones, flavonols and phenolic acids	pinobaskin, pinocembrin, hesperidin, naringin, naringenin, acacetin, apigenin, chrysin, luteolin, fisetin, galangin, isorhamnetin, kaempferol, quercetin, rutin, caffeic acid, gallic acid, 4-hydroxy-3-methoxyphenylethanol, 4-hydroxybenzoic ac-id-methyl ester, 4-hydroxybenzoic acid, 4-hydroxyhydrocinnamic acid	[88]
Vitamins	B1, B2, B3, B5, B6, B9, and provitamin A	[89]
Minerals	Cu, Fe, K, Mg, and Zn	[85]

**Table 5 pharmaceuticals-17-00646-t005:** Principal venom components.

Class	Substances	References
Carbohydrates	glucose, fructose	[95]
Proteins	mast cell degranulating peptide, melittin, secapine, tertiapine, apamin, procamine	[90,95]
Enzymes	phospholipase A2, hyaluronidase, phosphatase, glucosidase	[2,95]
Biogenic amines	Histamine, dopamine, noradrenaline	[2,95]
Amino acids	aminobutyric acid, α-amino acids	[95]

**Table 6 pharmaceuticals-17-00646-t006:** Tested cell line and hive product/component that showed efficacy.

Honeybee Product and/or Component	Cell Line	References
Honey	Human breast cancer cell line	[134]
	Cervical cell line	[134]
	MCF-7 (breast cancer cell line)	[128,131]
	PC-3 (prostate cell type)	[133]
Chrysin	Melanoma cell line	[135]
	Colon cancer cell line	[135]
	Brest cancer, prostate, cervical, liver, glioblastoma, lung and pancreas cell lines	[135,137]
Quercetin	Leukemic, breast cancer, human colon adenocarcinoma, prostate cancers, oral cancer, Ishiwa endometrial cancer, malignant mesothelioma	[138,144]
Propolis	Lung cancer cell line	[148]
	Human tongue squamous carcinoma	[151]
	Lymphocyte leukemia model	[149]
	Glioblastoma	[150]
Artepillin C	Gastrointestinal cell line	[158]
	Castration-resistant prostate cancer	[161]
Caffeic acid phnethyl ester	Oral squamous cell carcinoma and submucous fibrosis-derived fibroblasts	[164]
	Leukemic cell line	[165,167]
	Cervical cancer cell line	[166]
	Fibrosarcoma cell line	[163]
	Breast cancer	[168]
Pollen	Human prostate cancer, human lung carcinoma,	[185,186]
	Human bronchogenic carcinoma	[187]
Bee venom	Gastric cancer cell line	[196]
	Human malignant melanoma	[200]
	monkey kidney epithelial cells, colon adenocarcinoma cell line	[195]
Royal Jelly	Human breast cancer cell line	[210]
	Human hepatoma cell line	[211]
	Leukemia cell model	[207]

**Table 7 pharmaceuticals-17-00646-t007:** Hive product/component tested and pathogen/viral species toward which they demonstrated efficacy.

Honeybee Product and/or Component	Bacterial and Viral Species	References
Honey	*Staphylococcus aureus and Micrococcus luteus*	[231]
	*Staphylococcus aureus*, *Staphylococcus epidermidis*, *Enterobacter aerogenes*, *Salmonella enterica*, *Klebsiella pneumoniae*, *Escherichia coli*	[229]
	*Staphylococcus aureus* and *Helicobacter pylori*	[237]
	*Staphylococcus aureus*	[232]
	*Staphylococcus aureus* and *P. aeruginosa*	[239]
	Rubella virus and varicella zoster	[243]
	Herpes simplex virus	[244]
	Influenza virus	[245]
Pinocembrin and apigenin	*Streptococcus mutans*	[251]
Pinocembrin	*Staphylococcus aureus*, *Streptococcus mutans*, *Streptococcus sobrinus*, *Enterococcus faecalis*, *L. monocytogenes*, *Pseudomonas aeruginosa*, *Klebsiella pneumoniae*, *Proteus mirabilis*, *and Enterobacter aerogenes*	[257]
Cinnamic acid	*Streptococcus pyogenes*, *Bacillus* spp., *Mycobacterium tuberculosis*, *Aeromonas* spp. *Vibrio* spp., *Escherichia coli*, *Listeria monocytogenes*, *Micrococcus flavus*, *Pseudomonas aeruginosa*, *Salmonella enterica*, *Eterobacter clocae*, *Yersinia ruckeri*	[258,259,260]
Propolis	Influenza viruses	[14]
	Herpes virus	[13]
	HIV	[17]
	Newcastle disease virus	[19]
	Influenza viruses	[15]
	Poliovirus-type 1	[21]
	Human T-cell leukemia-lymphoma virus type 1	[18]
	RSV	[20]
	Dengue virus	[22]
	MRSA	[249]
	*Porphyromonas gingivalis*	[250]
	*Streptococcus mutans*	[251]
	*Staphylococcus aureus*, *Staphylococcus saprophyticus*, *Listeria monocytogenes and Enterococcus faecalis*	[252]
	*Candida glabrata*	[263]
	*Staphylococcus aureus*	[263]
	*Streptococcus pyogenes*	[267]
Bee venom	*Escherichia coli*, *Staphylococcus aureus*, *Borrelia burgdorferi*	[247,287,288]
	MRSA and *Enterococcus* spp.	[289]

**Table 8 pharmaceuticals-17-00646-t008:** Hive product/component tested and parasite species toward which they demonstrated efficacy.

Honeybee Product and/or Component	Parasite Species	Reference
Honey	*Leishmania tropica*	[298]
	*Plasmodium berghei*	[301]
Propolis	*Giardia duodenalis*	[299]
	*Toxocara vitulorum*	[300]
	*Schistosoma mansoni*	[324]
	*Leishmania tropica*	[298]
Bee venom	*Toxoplasma gondii*	[322,323]
	*Schistosoma mansoni*	[324]
Phospholipase A2	*Plasmodium falciparum*	[321]
	*Trypanosoma*	[320]

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
