# Peer review of "Hive Products: Composition, Pharmacological Properties, and Therapeutic Applications"

_pharmaceuticals, 2024, doi:10.3390/ph17050646_

Round 1

Reviewer 1 Report

Comments and Suggestions for Authors

The subject of the manuscript is interesting; however, in my opinion, the manuscript should be prepared with more attention. Some topics are discussed quite superficially. The tables are laconic and lack necessary details.

Detailed comments:

Tables should be completed. The authors should add the origin of the honey and appropriate references to the study in which the particular components were detected (and contents if provided). Furthermore, some components which was mention in the Section 4 (e.g. quinoline alkaloid

Tables are not cited in the text.

Table 4: some components mentioned as „Flavanones, flavones, flavonols and phenolic acids” do not belong to this class (e.g. octanoic acids ….)

Information provided in subsections of the fourth chapter should be presented in tabular form for the summarization of a particular type of activity. It could shortened the description in the text and make the manuscript more compact. This is typical for review papers and facilitates data tracking.

The authors should carefully review the abbreviations and ensure that all of them are explained when used for the first time

The same abbreviation was used for the different compounds, see line 373: phenethyl ester of caffeine (CAPE) and Line 419: “caffeic acid phenethyl ester 419 (CAPE)”

In Section 4.1 (Bee Products' Anticancer Properties), there is also information about antioxidant effects. They should be placed in a separate section. The role of ROS in anticancer effects is related to ROS stimulation, while antioxidant activity is responsible for the prevention of cancer development. These are different mechanisms of action and should be clearly stated.

Minor

Line 86, 142, 147, 184: lack of references regarding the chemical composition

Line 154-158: lack of references

Line 148: The range should be given from the lowest to the largest value.

Line 195: check the references order

Line 368: it should be: “Szliszka et al.”

“et al.'s” correct to “et al.”

Author Response

REVIEWER 1

The subject of the manuscript is interesting; however, in my opinion, the manuscript should be prepared with more attention. Some topics are discussed quite superficially. The tables are laconic and lack necessary details.

Response: we thank you for your appreciation of the manuscript. We have responded to your comments by highlighting the changes in the text.

Detailed comments:

Tables should be completed. The authors should add the origin of the honey and appropriate references to the study in which the particular components were detected (and contents if provided). Furthermore, some components which was mention in the Section 4 (e.g. quinoline alkaloid

R: a column with bibliographical references regarding honey, propolis, bee venom, pollen and royal jelly composition has been added in all tables.

Tables are not cited in the text.

R: thank you very much for your comment, we have cited the tables in the text as indicated

Table 4: some components mentioned as „Flavanones, flavones, flavonols and phenolic acids” do not belong to this class (e.g. octanoic acids ….)

R: we have deleted the components that had been inserted incorrectly, thank you for this important comment

Information provided in subsections of the fourth chapter should be presented in tabular form for the summarization of a particular type of activity. It could shortened the description in the text and make the manuscript more compact. This is typical for review papers and facilitates data tracking.

R: Thanks for your advice. We have added three summary tables

The authors should carefully review the abbreviations and ensure that all of them are explained when used for the first time

R: the abbreviations, when they were first written, were reported in extended form

The same abbreviation was used for the different compounds, see line 373: phenethyl ester of caffeine (CAPE) and Line 419: “caffeic acid phenethyl ester 419 (CAPE)”

R: thank you for bringing this typo to our attention. The term has been corrected

In Section 4.1 (Bee Products' Anticancer Properties), there is also information about antioxidant effects. They should be placed in a separate section. The role of ROS in anticancer effects is related to ROS stimulation, while antioxidant activity is responsible for the prevention of cancer development. These are different mechanisms of action and should be clearly stated.

R: the section was moved to the paragraph discussing the antioxidant properties of bee products.

Minor

Line 86, 142, 147, 184: lack of references regarding the chemical composition

R: thanks for the comment, the bibliographical references have been added

Line 154-158: lack of references

R: the bibliographical references have been added

Line 148: The range should be given from the lowest to the largest value.

R: now amended

Line 195: check the references order

R: thank you for highlighting this problem, a bibliographic reference was erroneously reported which we deleted

Line 368: it should be: “Szliszka et al.”

R: now amended

“et al.'s” correct to “et al.”

R: it has now been corrected throughout the text

Reviewer 2 Report

Comments and Suggestions for Authors

The manuscript, pharmaceuticals-2941959, reviews the components of the hive products, their pharmaceutical properties, and their practical applications.

The review is rather descriptive and does not evaluate the latest approaches that offer a more complex view of the pharmaceutical properties of the hive products by the complexity of these products.

For example, the antibacterial characteristics of the honey are influenced by its colloidal characteristics, the (macro)molecular crowding in the water pockets, and complex interactions.

  • Brudzynski, K., & Sjaarda, C. P. (2021). Colloidal structure of honey and its influence on antibacterial activity. Comprehensive Reviews in Food Science and Food Safety, 20(2), 2063-2080.
  • Tsavea, E., & Mossialos, D. (2019). Antibacterial activity of honeys produced in Mount Olympus area against nosocomial and foodborne pathogens is mainly attributed to hydrogen peroxide and proteinaceous compounds. Journal of Apicultural Research58(5), 756-763.
  • Brudzynski, K., Miotto, D., Kim, L., Sjaarda, C., Maldonado-Alvarez, L., & FukÅ›, H. (2017). Active macromolecules of honey form colloidal particles essential for honey antibacterial activity and hydrogen peroxide production. Scientific reports7(1), 7637.

Honey, due to natural deep eutectic solvent characteristics, increases the bioactivity of natural compounds.

  • Dimitriu, L., Constantinescu-Aruxandei, D., Preda, D., NichiÈ›ean, A. L., Nicolae, C. A., Faraon, V. A., ... & Oancea, F. (2022). Honey and its biomimetic deep eutectic solvent modulate the antioxidant activity of polyphenols. Antioxidants11(11), 2194.
  • Dai, Y., Jin, R., Verpoorte, R., Lam, W., Cheng, Y. C., Xiao, Y., ... & Chen, S. (2020). Natural deep eutectic characteristics of honey improve the bioactivity and safety of traditional medicines. Journal of ethnopharmacology250, 112460.

Propolis have a low bioavailability, which is increased by nanoformulation

  • Javed, S., Mangla, B., & Ahsan, W. (2022). From propolis to nanopropolis: An exemplary journey and a paradigm shift of a resinous substance produced by bees. Phytotherapy Research36(5), 2016-2041.

Pollen components bioavailability is increased by fermentation

  • Cheng, Y., Ang, B., Xue, C., Wang, Z., Yin, L., Wang, T., ... & He, Z. (2023). Insights into the fermentation potential of pollen: manufacturing, composition, health benefits, and applications in food production. Trends in Food Science & Technology, 104245.
  • UÈ›oiu, E., Matei, F., Toma, A., Diguță, C. F., Ștefan, L. M., Mănoiu, S., ... & Oancea, F. (2018). Bee collected pollen with enhanced health benefits, produced by fermentation with a Kombucha Consortium. Nutrients10(10), 1365.
  • Di Cagno, R., Filannino, P., Cantatore, V., & Gobbetti, M. (2019). Novel solid-state fermentation of bee-collected pollen emulating the natural fermentation process of bee bread. Food microbiology82, 218-230.

I believe mentioning such an approach should increase the review's attractiveness.

Author Response

REVIEWER 2

The manuscript, pharmaceuticals-2941959, reviews the components of the hive products, their pharmaceutical properties, and their practical applications. The review is rather descriptive and does not evaluate the latest approaches that offer a more complex view of the pharmaceutical properties of the hive products by the complexity of these products.

R: thank you for your review work, we have integrated all the suggestions you provided with your comments in the article, citing them. You'll find all integrations and modifications highlighted.

For example, the antibacterial characteristics of the honey are influenced by its colloidal characteristics, the (macro)molecular crowding in the water pockets, and complex interactions.

Brudzynski, K., & Sjaarda, C. P. (2021). Colloidal structure of honey and its influence on antibacterial activity. Comprehensive Reviews in Food Science and Food Safety, 20(2), 2063-2080.

Tsavea, E., & Mossialos, D. (2019). Antibacterial activity of honeys produced in Mount Olympus area against nosocomial and foodborne pathogens is mainly attributed to hydrogen peroxide and proteinaceous compounds. Journal of Apicultural Research, 58(5), 756-763.

Brudzynski, K., Miotto, D., Kim, L., Sjaarda, C., Maldonado-Alvarez, L., & FukÅ›, H. (2017). Active macromolecules of honey form colloidal particles essential for honey antibacterial activity and hydrogen peroxide production. Scientific reports, 7(1), 7637.

R: we have added the sentence you indicated and the bibliographical references

Honey, due to natural deep eutectic solvent characteristics, increases the bioactivity of natural compounds.

Dimitriu, L., Constantinescu-Aruxandei, D., Preda, D., Nichițean, A. L., Nicolae, C. A., Faraon, V. A., ... & Oancea, F. (2022). Honey and its biomimetic deep eutectic solvent modulate the antioxidant activity of polyphenols. Antioxidants, 11(11), 2194.

Dai, Y., Jin, R., Verpoorte, R., Lam, W., Cheng, Y. C., Xiao, Y., ... & Chen, S. (2020). Natural deep eutectic characteristics of honey improve the bioactivity and safety of traditional medicines. Journal of ethnopharmacology, 250, 112460.

R: we have added the sentence you indicated and the related bibliographical references

Propolis have a low bioavailability, which is increased by nanoformulation

Javed, S., Mangla, B., & Ahsan, W. (2022). From propolis to nanopropolis: An exemplary journey and a paradigm shift of a resinous substance produced by bees. Phytotherapy Research, 36(5), 2016-2041.

R: we have added the sentence you indicated and the bibliographical references in a novel section entitled “Future perspectives”

Pollen components bioavailability is increased by fermentation

Cheng, Y., Ang, B., Xue, C., Wang, Z., Yin, L., Wang, T., ... & He, Z. (2023). Insights into the fermentation potential of pollen: manufacturing, composition, health benefits, and applications in food production. Trends in Food Science & Technology, 104245.

Uțoiu, E., Matei, F., Toma, A., Diguță, C. F., Ștefan, L. M., Mănoiu, S., ... & Oancea, F. (2018). Bee collected pollen with enhanced health benefits, produced by fermentation with a Kombucha Consortium. Nutrients, 10(10), 1365.

Di Cagno, R., Filannino, P., Cantatore, V., & Gobbetti, M. (2019). Novel solid-state fermentation of bee-collected pollen emulating the natural fermentation process of bee bread. Food microbiology, 82, 218-230.

R: we have added the sentence you indicated and the bibliographical references in a novel section entitled “Future perspectives”

I believe mentioning such an approach should increase the review's attractiveness.

R: thank you very much for your revision work

Reviewer 3 Report

Comments and Suggestions for Authors

The present review aims to provide a thorough screening of the bioactive chemicals found in honeybee products and their beneficial biological effects. The manuscript may provide impetus to that branch of unconventional medicine that goes by the name of apitherapy. I found the paper to be overall well described with relevant information although must be improved attending the minor points for consideration for publishing in Pharmaceutics.

Major points:

1. Authors need to revise the spelling of the manuscript since it contains typos and grammar errors

2. In my opinion, the chemical structures of the discussed compounds must be included in the manuscript.

3. Authors must include a Conclusion section to make a strong statement about the "Future perspectives".

Comments on the Quality of English Language

Authors need to revise the spelling of the manuscript since it contains typos and grammar errors

Author Response

REVIEWER 3

The present review aims to provide a thorough screening of the bioactive chemicals found in honeybee products and their beneficial biological effects. The manuscript may provide impetus to that branch of unconventional medicine that goes by the name of apitherapy. I found the paper to be overall well described with relevant information although must be improved attending the minor points for consideration for publishing in Pharmaceutics.

R: thank you for your appreciation of our paper and for your important revision work. We have modified the document according to the indications you provided. Furthermore, we have responded point by point to your comments

Major points:

  1. Authors need to revise the spelling of the manuscript since it contains typos and grammar errors

R: thank you for your suggestion to help us improve the quality of the manuscript. The English has been revised.

  1. In my opinion, the chemical structures of the discussed compounds must be included in the manuscript.

R: we believe that such a section of added chemical structures would greatly burden an already extensively long and detailed manuscript. Furthermore, the compounds mentioned are numerous and of the most disparate classes. We would prefer, with your consent, to avoid adding a chemical structures section.

  1. Authors must include a Conclusion section to make a strong statement about the "Future perspectives".

R: thank you for your comment which helps us improve the overall quality of the manuscript. We have added a session entitled “Future perspectives” as you recommended

Reviewer 4 Report

Comments and Suggestions for Authors

The paper provides a good piece of work on hive products and their biological potential, the only problem is that the authors need to improve the writing style in some passages. Indeed, in some sections, certain sentences suffer from unclear phrasing or can be better elaborated, sometimes the sentences are too long or I feel a lack of finding the right terms. Moreover, as it is a narrative review, it is crucial that the authors maintain chronological order in their data compilation (This is my general opinion).

- More particularly, give a list of abbreviations: Ras-MAPK, PI3K-AKT, PLC-γ-CaM, and NF-kB, PtdIns(3,4)P2 and bvsPLA2, TBARS …..
- correct line 92: human herpesviruses
- correct lines 256-257: octanoic acids, 2-hexenedioic acid and its esters, dodecanoic acid and its ester, 1,2-benzenedicarboxylic acid, and benzoic acid, are not phenolics….

- Correct also in table 4: octanoic acids, 2-hexenedioic acid and its esters, dodecanoic acid and its ester, 1,2-benzenedicarboxylic acid, and benzoic acid are not phenolics
- Line 359 need space
- in italic in vitro, check throughout the text
- line 537; correct bee instead ee
- In Table 3, what is the distinction between vitamin P and flavonoids given that flavonoids are already mentioned in the table,
- correct line 249, is the 24-methylenecholesterol a fatty acid?
- rephrase lines 496-498
- rephrase lines 527-530 “Certain components of bee venom inhibit surface receptors by either dephosphorylating them or causing their degradation. This, in turn, modifies the signaling pathways downstream that are crucial for proliferation, metastasis, angiogenesis, and apoptosis (for example, the synergistic effect of BV sPLA2 and PtdIns(3,4)P2). “
- rephrase paragraph: 532-541; there are long sentences and there is redundancy, for example for Bcl2....
- Correct line 544
- lines 548, 551, 554: correct unit;
- lines 555 and 556, rephrase the sentence,
- lines 583-587: the sentence structure must be simplified and provide more context on how the findings fit into the understanding of royal jelly's anticancer activity in Ehrlich tumor mice
- lines 587-589: this paragraph could be written more simply: “It has been demonstrated that RJ inhibits the growth of MCF-7 human breast cancer cells, which was induced by the environmental estrogen bisphenol A, in another study conducted by Nakaya et al. (2007) [158]
- line 595, delete us
- minimize the text between 609-618
- lines 637-638: rephrase
- deletes lines 600-601: “We are aware that in vitro MTT tests were primarily used to evaluate the anticancer properties of the bee products
- line 771, need space
- 787, rephrase the sentence
- line 847: reference 254 does seem out of place in the antiparasitic potential section
- line 849: also for reference 256,
- line 858, reference 264 seems also not correct,
- lines 867-868, remove the sentence “It is thought that the phenolic compounds are what give Brazilian propolis its antioxidant properties
- line 934, this is not correct as phenylpropanoids are hydroxycinnamic derivatives which are a class of polyphenols,

- from the antioxidants section, I ask the authors to double-check the references until the end of the document,
- In lines 924-928, moving from one paragraph to another seems difficult, rephrase the paragraph, and say why flavonols can behave as prooxidants, and in what situations….
-The author also must review the polar paradox behavior, because they state that polar antioxidants are always better than non-polar ones, but this behavior depends strongly on the media used….check throughout the text and rephrase.
- homogenize the references, several are without doi

Comments on the Quality of English Language

Writing style can be improved

Author Response

REVIEWER 4

The paper provides a good piece of work on hive products and their biological potential, the only problem is that the authors need to improve the writing style in some passages. Indeed, in some sections, certain sentences suffer from unclear phrasing or can be better elaborated, sometimes the sentences are too long or I feel a lack of finding the right terms. Moreover, as it is a narrative review, it is crucial that the authors maintain chronological order in their data compilation (This is my general opinion).

R: thank you for your important review work which helps us improve the overall quality of the manuscript. The instructions you gave us were followed. The sentences you advised us to rephrase have been rephrased more adequately; the English language has been revisited. You can find your recommended changes underlined in the resubmitted manuscript. With regard to the chronological order of the narrative, we have taken care to include three tables showing by order of appearance in the scietific literature the findings discussed and cited in the manuscript. Rearranging events by annuality even in the text would be complicated because the manuscript is not purely narrative but there are also interspersed considerations that associate and elaborate on discoveries that occurred later to better explain a scientific finding that occurred earlier in time. With your permission, we would ask that the order currently set up be maintained in the text

- More particularly, give a list of abbreviations: Ras-MAPK, PI3K-AKT, PLC-γ-CaM, and NF-kB, PtdIns(3,4)P2 and bvsPLA2, TBARS …..

R: the abbreviations have been written out in full when they first appear, and a list of the abbreviations has been included.

- correct line 92: human herpesviruses

R: now amended

- correct lines 256-257: octanoic acids, 2-hexenedioic acid and its esters, dodecanoic acid and its ester, 1,2-benzenedicarboxylic acid, and benzoic acid, are not phenolics….

R: the indicated compounds have been deleted

- Correct also in table 4: octanoic acids, 2-hexenedioic acid and its esters, dodecanoic acid and its ester, 1,2-

benzenedicarboxylic acid, and benzoic acid are not phenolics

R: the indicated compounds have been deleted

- Line 359 need space

R: space has been added

- in italic in vitro, check throughout the text

R: checked throughout the text and corrected as suggested

- line 537; correct bee instead ee

R: now amended

- In Table 3, what is the distinction between vitamin P and flavonoids given that flavonoids are already mentioned in the table,

R: vitamin p has been deleted from the table

- correct line 249, is the 24-methylenecholesterol a fatty acid?

R: 24-methylenecholesterol was deleted

- rephrase lines 496-498

R: the sentence has been rephrased to make it more understandable.

- rephrase lines 527-530 “Certain components of bee venom inhibit surface receptors by either dephosphorylating them or causing their degradation. This, in turn, modifies the signaling pathways downstream that are crucial for proliferation, metastasis, angiogenesis, and apoptosis (for example, the synergistic effect of BV sPLA2 and PtdIns(3,4)P2). “

R: the sentence has been rephrased to make it more understandable.

- rephrase paragraph: 532-541; there are long sentences and there is redundancy, for example for Bcl2....

R: the sentence has been rephrased to make it more understandable.

- Correct line 544

R: the sentence has been rephrased to make it more understandable.

- lines 548, 551, 554: correct unit;

R: now amended

- lines 555 and 556, rephrase the sentence,

R: the sentence has been corrected and simplified

- lines 583-587: the sentence structure must be simplified and provide more context on how the findings fit into the understanding of royal jelly's anticancer activity in Ehrlich tumor mice

R: the sentence has been corrected and the concept has been explained better

- lines 587-589: this paragraph could be written more simply: “It has been demonstrated that RJ inhibits the growth of MCF-7 human breast cancer cells, which was induced by the environmental estrogen bisphenol A, in another study conducted by Nakaya et al. (2007) [158]”

R: the sentence has been simplified

- line 595, delete us

R: deleted

- minimize the text between 609-618

R: the text has been cut as per the suggestion

- lines 637-638: rephrase

R: the concept was expressed better

- deletes lines 600-601: “We are aware that in vitro MTT tests were primarily used to evaluate the anticancer properties of the bee products “

R: deleted

- line 771, need space

R: the space has been added

- 787, rephrase the sentence

R: the sentence has been reformulated

- line 847: reference 254 does seem out of place in the antiparasitic potential section

R: the reference “de Miranda, M.B.; Lanna, M.F.; Nascimento, A.L.B.; de Paula, C.A.; de Souza, M.E.; Felipetto, M.; da Silva Barcelos, L.; de Moura, S.A.L. Hydroalcoholic extract of Brazilian green propolis modulates inflammatory process in mice submitted to a low protein diet. Biomed. Pharmacother. 2019, 109, 610–620” has been deleted

- line 849: also for reference 256,

R: The reference “Yesmin, S.; Paul, A.; Naz, T.; Rahman, A.B.M.A.; Akhter, S.F.; Wahed, M.I.I.; Emran, T. Bin; Siddiqui, S.A. Membrane stabilization as a mechanism of the anti-inflammatory activity of ethanolic root extract of Choi (Piper chaba). Clin. Phytoscience 2020, 6, doi:10.1186/s40816-020-00207-7” has been deleted

- line 858, reference 264 seems also not correct,

R: the indicated bibliographic reference has been replaced with: Banskota, A. H., Tezuka, Y., & Kadota, S. (2001). Recent progress in pharmacological research of propolis. Phytotherapy research, 15(7), 561-571.

- lines 867-868, remove the sentence “It is thought that the phenolic compounds are what give Brazilian propolis its antioxidant properties

R: deleted

- line 934, this is not correct as phenylpropanoids are hydroxycinnamic derivatives which are a class of polyphenols,

R: the sentence was correct in accordance with the suggestion

- from the antioxidants section, I ask the authors to double-check the references until the end of the document,

R: checked

- In lines 924-928, moving from one paragraph to another seems difficult, rephrase the paragraph, and say why flavonols can behave as prooxidants, and in what situations….

R: to better explain the concept, a phrase between line 991 and 994 and an associated bibliographic note was added.

-The author also must review the polar paradox behavior, because they state that polar antioxidants are always better than non-polar ones, but this behavior depends strongly on the media used….check throughout the text and rephrase.

R: a sentence on the polar paradox with related citation was added to line 921

- homogenize the references, several are without doi

R: we ask the reviewer for time before complying with this review advice. The bibliographical references were added using software that simplifies the management of the bibliography. The doi should now be added manually. However, every time a bibliographic note is added, the manual additions are deleted. We ask the reviewer to wait until there is agreement for the publication of the manuscript. Before publication, when we are sure that we will not have to add other bibliographic notes via software, we will add the doi

Round 2

Reviewer 1 Report

Comments and Suggestions for Authors

Manuscript has been improved. However, I have still a few minor suggestions.

What does it mean „~” symbol (see lines 153, 154)?

Table 1: I haven't encountered placing abbreviations in the main body of the manuscript. Usually, they are placed at the beginning or end of the text. 

The formatting of the tables should be in accordance with the journal's requirements.

Table 3, line 3: Which of the listed compounds belong to terpenoids? Additionally, the chemical composition of propolis is described based on only one paper from 2014 (review paper). There are many more recent papers describing this issue. Moreover, I think that authors should base their work on sourced papers (not on review papers).

Tables 7-9: References should be given  as a number (the author's name is unnecessary).

Author Response

Reviewer 1

Manuscript has been improved. However, I have still a few minor suggestions.

What does it mean „~” symbol (see lines 153, 154)?

Response: thanks for pointing out this typo. the incorrectly entered symbol has been replaced with ±

Table 1: I haven't encountered placing abbreviations in the main body of the manuscript. Usually, they are placed at the beginning or end of the text.

R: many thanks for this comment, as per the suggestion we moved the table to the end of the manuscript

The formatting of the tables should be in accordance with the journal's requirements.

R: we have changed the formatting of the tables according to your suggestion.

Table 3, line 3: Which of the listed compounds belong to terpenoids? Additionally, the chemical composition of propolis is described based on only one paper from 2014 (review paper). There are many more recent papers describing this issue. Moreover, I think that authors should base their work on sourced papers (not on review papers).

R: we have added a row for terpenoids, furthermore, as suggested, we have cited research articles to support the compounds mentioned in the table

Tables 7-9: References should be given as a number (the author's name is unnecessary).

R: as suggested, the authors' names have been deleted from the column

Reviewer 4 Report

Comments and Suggestions for Authors

I believe that the antiinflammatory effects section should be just before or just after the anticancer potential section

Lines 153-154, check if symbols are corrects

Line 154, and from

In table 3 , there is no terpenoids, only phenolic acids , hydroxycinnamic acids and flavonoids, remove terpenoids, I suggest to use polyphenolics in place ;

Line 584 :p53 instead P53,

Line 584 replace by ‘’which promotes cell cycle arrest, instead ''with consequent arrest and death of the cell cycle !!!

Bcl-2 instead BCL-2, check thoughout the text

Line 602 : need space,

Rephrase lines 611-613 by ‘’ Melittin’s potential anticancer effect may be due to its ability to initiate the apoptotic pathway through the release of cytochrome-c, leading to the activation of caspase-9, which in turn triggers caspase-3

Rephrase the paragraph 613-615 : Further research on this topic revealed that melittin interferes with the formation of tumor metastases by inhibiting F-actin rearrangement and activation of theprimarily inhibits F-actin rearrangement and epidermal growth factor receptor (EGFR) activation, which in turn stops melanoma cells from

The paragraph 617-619 is very poorly expressed!!! to be rephrased

Delete the line 625

Write the scientific names including bacterial strains in full, check the entire document

Line 730-732 : last paragraph poorly expressed,

Paragraph 817- 818 : rephrase

Lines 897-904, delete the unnecessary paragraph ‘’The absence of a reliable and secure treatment is one of the factors causing these illnesses to spread. According to reports, the current pharmacotherapy options have serious drawbacks, in- cluding being less effective than ideal—especially when it comes to treating a particular form of the parasite—having variable efficacy rates, having unpleasant side effects, requiring lengthy treatment or administration periods, and some parasites developing resistance to their effects [252–254]. In light of this situation, there is a great need to discover and develop novel, effective antiparasitic treatments that are reasonably priced and have few side effects.

Line 941 Plasmodium whole word in italic

Line 965 : according to Zhang et al."

Lines 1054-1057, is the paragraph useful in relation to the review scope ?

Replace the pargraph line 1061 by ‘’Another aspect influencing these qualities is the time of harvest ,

In lines 1133-1137, replace hyphens with commas

Line 1335, I am not aware of any flavonoid called "chyrus," plus you do not provide a reference, so this line must be deleted

Rephrase lines 1336-1340

Comments on the Quality of English Language

English must be reviewed

Author Response

Reviewer 4

I believe that the antiinflammatory effects section should be just before or just after the anticancer potential section

Response: thanks for the advice, we moved the section before the indicated paragraph

Lines 153-154, check if symbols are corrects

R: now amended

Line 154, and from

R: corrected as suggested

In table 3 , there is no terpenoids, only phenolic acids , hydroxycinnamic acids and flavonoids, remove terpenoids, I suggest to use polyphenolics in place ;

R: many thanks for this comment, we have created a specific row for terpenoids in the table

Line 584 :p53 instead P53,

R: now amended

Line 584 replace by ‘’which promotes cell cycle arrest, instead ''with consequent arrest and death of the cell cycle !!!

R: we corrected as suggested

Bcl-2 instead BCL-2, check thoughout the text

R: we corrected as suggested

Line 602 : need space,

R: now amended

Rephrase lines 611-613 by ‘’ Melittin’s potential anticancer effect may be due to its ability to initiate the apoptotic pathway through the release of cytochrome-c, leading to the activation of caspase-9, which in turn triggers caspase-3

R: many thanks for this comment, we have rephrased the period

Rephrase the paragraph 613-615 : Further research on this topic revealed that melittin interferes with the formation of tumor metastases by inhibiting F-actin rearrangement and activation of theprimarily inhibits F-actin rearrangement and epidermal growth factor receptor (EGFR) activation, which in turn stops melanoma cells from

R: we have rephrased the period as per the suggestion

The paragraph 617-619 is very poorly expressed!!! to be rephrased

R: thanks for this advice, we have rephrased the period

Delete the line 625

R: we have deleted the indicated sentence

Write the scientific names including bacterial strains in full, check the entire document

R: we have modified as suggested

Line 730-732 : last paragraph poorly expressed,

R: we have rephrased the period

Paragraph 817- 818 : rephrase

Lines 897-904, delete the unnecessary paragraph ‘’The absence of a reliable and secure treatment is one of the factors causing these illnesses to spread. According to reports, the current pharmacotherapy options have serious drawbacks, in- cluding being less effective than ideal—especially when it comes to treating a particular form of the parasite—having variable efficacy rates, having unpleasant side effects, requiring lengthy treatment or administration periods, and some parasites developing resistance to their effects [252–254]. In light of this situation, there is a great need to discover and develop novel, effective antiparasitic treatments that are reasonably priced and have few side effects.

R: we have rephrased the period

Line 941 Plasmodium whole word in italic

R: now amended

Line 965 : according to Zhang et al."

R: we have modified as suggested

Lines 1054-1057, is the paragraph useful in relation to the review scope ?

R: we have deleted it

Replace the pargraph line 1061 by ‘’Another aspect influencing these qualities is the time of harvest

R: we have modified as suggested

In lines 1133-1137, replace hyphens with commas

R: we have inserted the sentences indicated in brackets

Line 1335, I am not aware of any flavonoid called "chyrus," plus you do not provide a reference, so this line must be deleted

R: we have deleted the sentence

Rephrase lines 1336-1340

R: we rephrased the period as suggested
